# Marine Pharmacology in 2014–2015: Marine Compounds with Antibacterial, Antidiabetic, Antifungal, Anti-Inflammatory, Antiprotozoal, Antituberculosis, Antiviral, and Anthelmintic Activities; Affecting the Immune and Nervous Systems, and Other Miscellaneous Mechanisms of Action

**DOI:** 10.3390/md18010005

**Published:** 2019-12-19

**Authors:** Alejandro M. S. Mayer, Aimee J. Guerrero, Abimael D. Rodríguez, Orazio Taglialatela-Scafati, Fumiaki Nakamura, Nobuhiro Fusetani

**Affiliations:** 1Department of Pharmacology, College of Graduate Studies, Midwestern University, 555 31st Street, Downers Grove, IL 60515, USA; aguerrero89@midwestern.edu; 2Molecular Sciences Research Center, University of Puerto Rico, 1390 Ponce de León Avenue, San Juan, PR 00926, USA; abimael.rodriguez1@upr.edu; 3Department of Pharmacy, University of Naples “Federico II”, Via D. Montesano 49, I-80131 Napoli, Italy; scatagli@unina.it; 4Department of Chemistry and Biochemistry, Graduate School of Advanced Science and Engineering, Waseda University, 3-4-1 Okubo, Shinjuku-ku, Tokyo 169-8555, Japan; what-will_be.x2@akane.waseda.jp; 5Fisheries and Oceans Hakodate, Hakodate 041-8611, Japan; anobu@fish.hokudai.ac.jp

**Keywords:** drug, marine, chemical, metabolite, natural product, pharmacology, pharmaceutical, review, toxicology, pipeline

## Abstract

The systematic review of the marine pharmacology literature from 2014 to 2015 was completed in a manner consistent with the 1998–2013 reviews of this series. Research in marine pharmacology during 2014–2015, which was reported by investigators in 43 countries, described novel findings on the preclinical pharmacology of 301 marine compounds. These observations included antibacterial, antifungal, antiprotozoal, antituberculosis, antiviral, and anthelmintic pharmacological activities for 133 marine natural products, 85 marine compounds with antidiabetic, and anti-inflammatory activities, as well as those that affected the immune and nervous system, and 83 marine compounds that displayed miscellaneous mechanisms of action, and may probably contribute to novel pharmacological classes upon further research. Thus, in 2014–2015, the *preclinical* marine natural product pharmacology pipeline provided novel pharmacology as well as new lead compounds for the *clinical* marine pharmaceutical pipeline, and thus continued to contribute to ongoing global research for alternative therapeutic approaches to many disease categories.

## 1. Introduction

The aim of the present review is to consolidate 2014–2015 *preclinical* marine pharmacology, with a format similar to the previous nine reviews of this series, which cover the period 1998–2013 [1,2,3,4,5,6,7,8,9]. The peer-reviewed articles were retrieved from searches in the following databases: MarinLit, PubMed, Chemical Abstracts^®^, ISI Web of Knowledge, and Google Scholar. As in our previous work, we have limited our review to include bioactivity and/or pharmacology of structurally characterized marine chemicals, which we have classified using a modification of Schmitz’s chemical classification [10] into six major chemical classes, namely, polyketides, terpenes, peptides, alkaloids, shikimates, and sugars. The preclinical antibacterial, antifungal, antiprotozoal, antituberculosis, antiviral, and anthelmintic pharmacology of marine chemicals is reported in Table 1, with the structures shown in Figure 1. Marine compounds that affected the immune and nervous systems, with antidiabetic and anti-inflammatory effects, are exhibited in Table 2, with their respective structures consolidated in Figure 2. Finally, marine compounds affecting a variety of cellular and molecular targets are noted in Table 3, and their structures are shown in Figure 3. 

Several publications during 2014–2015 reported on several marine extracts or structurally uncharacterized compounds, with novel *preclinical* and/or *clinical* pharmacology: in vitro phenotype-guided the discovery of natural products using cytological profiling to predict modes of action of bioactive constituents in extracts [11]; Colombian and Brazilian marine organisms as source of *antibacterial* extracts with bacterial quorum sensing inhibitory activity [12]; a first report describing antimicrobial activity in extracts from cultivable fungi associated with Antarctic marine sponges [13]; an extensive study on the antimicrobial activity of crude extracts from several species of red algae from Madagascar [14]; bioactive compounds along with a purified macrolactin with broad spectrum antibacterial activity in crude extracts from *B. subtilis* MTCC10403 associated with the Indian brown seaweed *A. longifolius* [15]; potential β-lactamase inhibitory activity in a Bay of Bengal marine *Streptomyces* sp. PM49 with antibacterial properties on multidrug-resistant pathogens [16]; in vitro *anti-inflammatory* activity of an extract and individual components in the spiny seastar *M. glacialis* that inhibited “different levels of the inflammation pathway” [17]; in vitro anti-inflammatory activity of several galactolipids isolated from methanol extracts of cultivated red alga *Chondrus crispus* that were proposed “to counter inflammation associated with NO-mediated disorders” [18]; in vitro immunomodulatory activity of an extract of the Bohai sea ascidian *Styela clava* that exhibited proliferative activity and promoted nitric oxide (NO) release from mouse lymphocytes and macrophages [19]; in vivo anti-inflammatory activity of a new nucleoside, dragmacidoside isolated from an extract of a marine Red sea sponge *Dragmacidon coccinea* [20]; in vivo anti-inflammatory and analgesic activities of an organic extract and its semipurified fractions from the Tunisian gorgonian *Eunicella singularis*, which suggested that “components of the active fraction can be used to treat various anti-inflammatory diseases” [21]; in vitro anti-inflammatory activity of 30 compounds in a methanol extract of the digestive gland of the Mediterranean sea hare *Aplysia depilans,* which could provide “beneficial anti-inflammatory effects” [22]; and, as part of anti-obesity nutraceuticals research, the anti-adipogenic activity of phlorotannins isolated from the edible brown alga *Ecklonia cava* were reported to be a “promising source for utilization against obesity and related complications” [23].

## 2. Marine Compounds with Antibacterial, Antifungal, Antiprotozoal, Antituberculosis, Antiviral, and Anthelmintic Activities

Table 1 presents 2014–2015 preclinical pharmacological research on the antibacterial, antifungal, antiprotozoal, antituberculosis, antiviral, and anthelmintic activities of marine natural products (**1**–**133**) shown in Figure 1.

### 2.1. Antibacterial Activity

During 2014–2015, 48 studies reported *antibacterial* marine natural products (**1**–**64**) isolated from bacteria, fungi, tunicates, sponges, soft corals, sea snakes, fish, and algae; a research enterprise focused on the discovery of novel chemical leads to treat emerging drug-resistant bacterial infections.

As shown in Table 1 and Figure 1, nine publications reported on the mode of action of marine-derived antibacterial compounds. Rodríguez and colleagues reported on “a practical synthesis of the axinellamines” (**1**, **2**), as well as their broad spectrum Gram-positive and Gram-negative antibacterial activity, probably resulting from “secondary membrane destabilization…consistent with the inhibition of normal septum formation” [24]. Moon and colleagues characterized a new pentacyclic antibiotic, buanmycin (**3**), isolated from a Korean marine *Streptomyces* strain, which was active towards Gram-native *Salmonella enterica* that causes salmonellosis, by inhibiting sortase A, an enzyme involved in bacterial adhesion and proposed as a “promising target for antibiotic discovery” [25]. Wei and colleagues discovered a novel peptide cathelicidin (**4**) from the Chinese sea snake *Hydrophis cyanocinctus* with potent antimicrobial activity against 35 strains of 48 human pathogenic bacteria, probably by a mechanism that involved “disruption of cell membrane and lysis of bacterial cells…resulting in cellular disruption of both Gram-positive and Gram-negative bacteria” [26]. Silva and colleagues demonstrated that the antimicrobial peptide clavanin A (**5**) significantly reduced *E. coli* and *S. aureus*-infected mice mortality with concomitant reduction of proinflammatory cytokines, thus proposing that clavalin A “… will facilitate studies on the development of novel peptide-based strategies for the treatment of infected wounds and sepsis” [27]. Abdelmohsen and colleagues investigated the new sterol gelliusterol E (**6**) from the Red sea sponge *Callyspongia* aff. *implexa* and showed that it inhibited both the primary infection by *Chlamydia trachomatis*, an obligate intracellular Gram-negative bacterium, as well as the production of viable progeny, and thus the developmental cycle of this bacterium [28]. Pieri and colleagues described new ianthelliformisamine B and C (**7, 8**) from the marine sponge *Suberea ianthelliformis* as antibiotic enhancers against resistant Gram-negative bacteria by a mechanism described as “altered proton homeostasis”, and thus probably affecting drug transport [29]. Huang and colleagues showed that the antimicrobial peptide pardaxin (**9**) isolated from the Red sea flatfish *Pardachirus marmoratus* protected mice from a lethal dose of methicillin-resistant *Staphylococcus aureus*, while also accelerating wound healing, increasing monocytes’ and macrophages’ recruitment, as well as expression of vascular endothelial growth factor [30]. Eom and colleagues described the mechanism of antibacterial activity of the phlorotannin phlorofucofuroeckol-A (**10**) isolated from the edible brown alga *Eisenia bicyclis*, which was shown to involve suppression of several *mec* operon genes in methicillin-resistant *Staphylococcus aureus* as well as the production of penicillin-binding protein 2a, considered as the “primary cause of methicillin resistance” [31]. Hassan and colleagues reported a new depsipeptide salinamide F (**11**), isolated from a marine-derived *Streptomyces* sp. strain CNB-091 that was observed to significantly inhibit RNA polymerase (RNAP) from both Gram-positive and Gram-negative bacteria, but “does not interact with the rifampin binding site on RNAP” [32].

As shown in Table 1 and Figure 1, 53 marine natural products (**12**–**64**), some of them novel, were reported to exhibit antibacterial activity with MICs <10 μg/mL or 10 μM against several Gram-positive and Gram-negative bacterial strains, although the mechanism of action for these compounds remained undetermined: two antimicrobial peptides, piscidins 3 and 4 (**12, 13**) originally isolated from the fish tilapia *Oreochromis niloticus* [33]; a new bisthiodiketopiperazine adametizine A (**14**), isolated from a marine sponge-derived fungus *Penicillium adametzioides* AS-53 [34]; two dimeric bromopyrrole alkaloids agelamadins A and B (**15, 16**), isolated from the Okinawan marine sponge *Agelas* sp. [35]; a butyrolactone derivative (**17**), isolated from the fermentation broth of a South China sea gorgonian *Muricella abnormalis*-derived *Aspergillus* sp. XS-20090B15 fungus [36]; a meroditerpene aszonapyrone A (**18**), isolated from the Thai marine sponge *Chondrilla asutraliensis*-associated fungus *Neosartorya paulistensis* (KUFC 7897) [37]; a new meroterpenoid, austalide R (**19**), from a fungus *Aspergillus* sp. isolated from the Mediterranean sponge *Tethya aurantium* [38]; a novel citrifelin B (**20**) with a unique tetracyclic framework was characterized from a co-culture of marine-derived fungi *Penicillium citrinum* and *Beauveria felina* [39]; a novel cyclohexadepsipeptide desmethylisaridin C1 (**21**) identified from a marine bryozoan-derived fungus *Beauveria felina* EN-135 [40]; two novel polybrominated diphenyl ethers (**22, 23**), isolated from the cosmopolitan marine sponge *Dysidea* spp. [41]; a new pentacyclic cytochalasin diaporthalasin (**24**), isolated from the marine-derived fungus *Diaporthaceae* sp. PSU-SP2/4 [42]; natural brominated furanones (**25, 26**), isolated from the marine alga *Delisea pulchra* [43]; a novel meroterpenoid aureol B (**27**), isolated from the Micronesian *Dysidea* sp. sponge [44]; a novel meroterpenoid dysidinoid A (**28**), isolated from the South China sea sponge *Dysidea* sp. [45]; a terpenoid fuscoside E peracetate (**29**) and a lipid batyl alcohol (**30**), isolated from the Colombian soft coral *Eunicea* sp. [46]; a new aromatic butyrolactone flavipesin A (**31**), isolated from a marine-derived endophytic fungus *Aspergillus flavipes* [47]; new non-cytotoxic lipopeptides gageopeptides A–D (**32, 33, 34, 35**) [48] and gageotetrins A–C (**36, 37, 38**) [49], isolated from a Korean marine-derived *Bacillus subtilis* strain 109GGC020; a new cyclic depsipeptide hormaomycin B (**39**), isolated from a Korean marine mudflat-derived actinomycete *Streptomyces* sp. strain SNM55 [50]; a new glycolipid ieodoglucomide C (**40**) from the Korean marine-derived bacterium *Bacillus licheniformis* strain 09IDYM23 [51]; a new polycyclic tetramic acid macrolactam isoikarugamycin (**41**) from Equatorial Guinean *Streptomyces zhaozhouensis* strain CA-185989 [52]; a bromopyrrole alkaloid keramadine (**42**), isolated from the Okinawan marine sponge *Agelas* sp. [53]; “unprecedented” 9,11 secosterols with “the 2-ene-1,4-dione moiety” (**43, 44**) from the Korean marine sponge *Ircinia* sp. [54,55]; two polybrominated diphenyl ethers (**45, 46**) from the Papuan New Guinea marine sponge *Lendenfeldia dendyi* and the soft coral *Sinularia dura* [56]; a novel polyketide lindgomycin (**47**) from the Baltic sea and Arctic Lindgomycetaceae family marine fungal strains KF970 and LF327 [57]; a novel cycloheptadepsipeptide marfomycin D (**48**)**,** isolated from a South China marine *Streptomyces drozdowiczii* SCSIO 10141 [58]; an glycohexadepsipeptide-polyketide mollemycin A (**49**) from an Australian marine-derived *Streptomyces* sp. strain CMB-M0244 [59]; a novel laurene-type sesquiterpene neolaurene (**50**) from a Bornean marine alga *Laurencia nangii* [60]; a new polyketide and ambuic acid analogue penicyclone A (**51**), isolated form an extract from a deep-sea derived fungus *Penicillium* sp. F23-2 [61]; a new phenolic enamide (**52**), characterized from the Chinese marine alga *Codium fragile*-derived endophytic fungus *Penicillium oxalicum* strain EN-290 [62]; a novel meroterpenoid puupehenol (**53**), isolated from a Hawaiian sponge *Dactylospongia* sp. [63]; a new scalarane sesterpene phyllospongin E (**54**), isolated from the Egyptian Red sea sponge *Phyllospongia lamellosa* [64]; two new rare pyrane-based cembranoids sarcotrocheliol acetate (**55**) and sarcotrocheliol (**56**), isolated from the Red sea soft coral *Sarcophyton trocheliophorum* [65]; a novel depsidone-based analogue spiromastixone J (**57**), isolated from the fermentation broth of a deep-sea *Spiromastix* sp. fungus [66]; a new spirocyclic drimane stachyin B (**58**), identified in the mycelia and culture broth of a North sea *Stachybotrys* sp. fungus strain MF347 [67]; a naphthacene glycoside SF2446A2 (**59**), isolated from a culture of *Streptomyces* sp. strain RV15 derived from the Mediterranean sponge *Dysidea tupha* [68]; three new subergosterones A–C (**60**–**62**), obtained from the South China sea gorgonian coral *Subergorgia rubra* [69]; an amino-polyketide vitroprocine A (**63**), isolated from the marine bacterium *Vibrio* sp. strain QWI-06 [70]; and a new polyacetylene derivative xestospongiamide (**64**), isolated from the Red sea sponge *Xestospongia* sp. [71].

Furthermore, during 2014–2015, several other marine natural products, some of them novel, reported antimicrobial activity in MICs or IC_50_’s ranging from 10 to 50 μg/mL, or 10–50 μM, respectively, and thus, because of their lower antibacterial potency, were *excluded* from Table 1 and Figure 1: tetracyclic sesterterpenes from a Korean marine sponge *Clathria gombawuiensis* sp. (MIC = 6.25–25 μg/mL) [129]; the antimicrobial peptide isolated from the mucus of the hagfish *Myxine glutinosa* and several structural analogs (MICs = 1.2–50 µM) [130]; bromopyrrole alkaloids agelamadins C–E (IC_50_ = 32 μg/mL) from the Okinawan marine sponge *Agelas* sp. [131]; a new indole diterpenoid from the fungus *Aspergillus flavus* (MIC = 20.5 µM), isolated from the Chinese prawn *Penaeus vannamei* [132]; two new bromopyrrole alkaloids isolated from the Okinawan sponge *Agelas* sp., namely 2-debromomonagelamide U and 2-debromomukanadin G (MIC=32 μg/mL) [133]; seven known and one new sesquiterpene named epoxysubergorgic acid isolated from the Chinese gorgonian coral *S. suberosa* (MIC>8 μg/mL) [134]; a new bromotyrosine alkaloid aplysamine 8 (MIC=31 μg/mL) from an Australian marine sponge *Pseudoceratina purpurea* [135], alternariol derivatives (MIC = 50 μg/disk) from the endophytic fungus *Alternaria alternata* isolated from the Red sea soft coral *Litophyton arboretum* [136]; new polyketides amphidins C–F (MIC = 16 and 32 μg/mL), isolated from the culture broth of dinoflagellate *Amphidinium* sp. [137]; a new 1-deoxysphingoid, 3-*epi*-xestoaminol C (MIC = 32.6 µM), isolated from the New Zealand brown alga *Xiphophora chondrophylla* [138]; a new cyclic pentapeptide, asperpeptide A (MIC = 12.5 µM), isolated from the gorgonian-derived fungus *Aspergillus* sp. [139]; a nucleoside derivative, kipukasin H (MIC = 12.5 µM), isolated from the fungus *A. versicolor* derived from the Xisha islands, South China sea gorgonian *D. gemmacea* [140]; a new *O*-containing heterocyclic compound named felinone B (MIC = 32 μg/mL) from an extract of *B. feline* EN-135, a fungus isolated from an unidentified marine bryozoan [141]; new xanthone microluside A (MIC = 10–13 µM) from a Red sea marine sponge *S. vagabunda*-derived *Micrococcus* sp. EG45 [142]; a new cyclohexapeptide desotamide B (MIC = 12–16 μg/mL) from a South China sea marine microbe *S. scopuliridis* SCSIO ZJ46 [143]; new linear lipopeptides (MIC = 16 and 32 μg/mL) from a Korean marine *Bacillus subtillis* [144]; a new streptophenazine K (MIC = 14.5–21.6 µM) from bacteria isolated from a Baltic sea sponge *Halichondria panacea* [145]; a novel polyketide amantelide A (MIC = 32 µM) from a Guamanian Oscillatoriales cyanobacterium [146]; a novel echinomycin analog, quinomycin G (MIC = 16–64 μg/mL), isolated from *Streptomyces* sp. LS298 from a Hainan marine sponge *Gelliodes carnosa* [147]; and new cyclic lipopeptides gageopeptins A and B (MIC = 6 and 32 μg/mL) from a marine-derived strain *Bacillus* sp. 109GGC020 [144].

### 2.2. Antifungal Activity

Sixteen studies during 2014–2015 reported on the *antifungal* activity of several marine natural products (**64**–**83**) isolated from marine bacteria, dinoflagellates, sponges, sea cucumbers, and algae, a slight increase from our last review [9] and previous reviews of this marine pharmacology series.

As shown in Table 1 and Figure 1, three reports described antifungal marine chemicals with novel mechanisms of action. Lee and colleagues investigated the new macrocyclic lactone antifungal bahamaolide A (**65**) isolated from the culture of marine actinomycete *Streptomyces* sp. CNQ343 [72]. Detailed studies determined that the compound inhibited isocitrate lyase (ICL) mRNA expression, suggesting it might be used for treatment of “*C. albicans* infections via inhibition of ICL activity”. Sugiyama and colleagues characterized the biological activity of the polyene macrolactam heronamide C (**66**) isolated from a marine-derived *Streptomyces* sp. [73]. The heronamide C was shown to induce abnormal cell wall morphology by “perturbing membrane microdomains”. Wyche and colleagues reported a novel marine-derived polyketide forazoline A (**67**) isolated from an *Actinomadura* sp. strain WMMB-499 cultivated from the ascidian *Ecteinascidia turbinata* [74]. Using chemical genomics, the authors proposed forazoline A worked in vivo in mice against the fungus *Candida albicans* by affecting cell membrane integrity by a “novel mechanism of action from known antifungal agents”.

As shown in Table 1 and Figure 1, several marine natural products showed antifungal activity with MICs that were either less than 10 μg/mL, 10 μM, or 10 μg/disk, but no mechanism of action studies were reported in the papers: an alkaloid aaptamine derivative (**68**), isolated from the South China sea sponge *Aaptos aaptos* [75]; a new linear polyketide amphidinin G (**69**), isolated from a Japanese symbiotic marine dinoflagellate *Amphidinium* sp. discovered in a marine flatworm *Amphiscolops* sp. [76]; a new polyketide amphidinol 18 (**70**), isolated from the dinoflagellate *Amphidinium carterae* strain CCMP121 [77]; new crambescin A2 (**71**–**73**), alkaloid homologues from the Bahamian marine sponge *Pseudaxinella reticulata* [78]; two new saponins cousteside C (**74**) and D (**75**), reported from the Egyptian Red sea cucumber *Bohadschia cousteaui* [79]; two new laurane-type sesquiterpenes, laurepoxyene (**76**) and 3β-hydroxyperoxyaplysin (**77**), and a new polyunsaturated fatty acid ethyl ester (**78**), isolated from the Chinese red alga *Laurencia okamurai* [80,81]; a novel dilactone-tethered pseudo-dimeric peptide mohangamide A (**79**), isolated from a Korean marine *Streptomyces* sp. [82]; a novel pleosporallin E (**80**), isolated from a marine fungus *Pleosporales* sp., discovered on the South China sea alga *Enteromorpha clathrata* [83]; a lysophospholipid (**81**), isolated from the South China sea sponge *Spirastrella purpurea* [84]; an acetylenic fatty acid derivative taurospongin A (**82**) from an Okinawan marine sponge SS-1202, family Spongiidae [85], and a new non-sulphated triterpene glycoside variegatuside D (**83**) from the south China sea cucumber *Stichopus variegates* [86]. Mechanism of action studies will be required to characterize the antifungal pharmacology of these marine-derived natural compounds.

In addition, novel structurally-characterized marine molecules with antifungal MICs or IC_50_’s greater than 10 μg/mL, 10 μM, or 10 μg/disk, which have been *excluded* from Table 1 and Figure 1 because of their weaker bioactivity: a new bromopyrrole alkaloid mukanadine G (IC_50_ = 8–16 μg/mL) isolated from the Okinawan marine sponge *Agelas* sp. [53]; a new C24-acetylenic acid, biemnic acid (MIC = 100 μg/disk) isolated from the Red sea sponge *Biemna ehrenbergi* [148]; two sulfated steroid-aminoacid conjugates isolated from the Irish marine sponge *Polymastia boletiformis* (MIC = 100 μg/disk) against *C. albicans* [149]; two novel lysophospholids from the Guanxi sponge *S. purpurea* (IC_50_ = 16 and 32 μg/mL) [84]; two new bromotyrosine alkaloids, tyrokeradine G and H, isolated from an Okinawan Verongid marine sponge (MIC = 16 and 32 μg/mL) [150]; two highly brominated polyphenols isolated from the Qingdao red alga *S. latiuscula* (MIC = 12.5, 25 μg/mL) [151]; one novel anhydride metabolite, tubingenoic anhydride A (MIC = 330 μM), from the Mediterranean fungus *A. tubingensis* (Strain OY907) [152]; and a novel compound terretrione D (MIC = 32 μg/mL) from a tunicate-derived fungus *Penicillium* sp. CYE-97 [153]. These novel marine compounds may contribute to the antifungal preclinical and clinical pipeline upon further research. 

### 2.3. Antiprotozoal and Antituberculosis Activity

As shown in Table 1, during 2014–2015, twenty-four studies contributed to novel findings on *antiprotozoal (antimalarial, antileishmanial, and antitrypanosomal)* and *antituberculosis* pharmacology of structurally characterized marine natural products (**84**–**112**), very similar to our previous 1998–2013 marine pharmacology reviews [1,2,3,4,5,6,7,8,9].

Malaria, a global disease caused by protozoan genus *Plasmodium* (*P. falciparum*, *P. ovale*, *P. vivax* and *P. malariae*), currently affects over 2 billion people worldwide. Contributing to the global search for novel antimalarial drugs, and as presented in Table 1, 11 marine molecules (**84**–**94**) isolated from bacteria, molluscs, sponges, and soft corals were shown during 2014–2015 to possess *antimalarial activity*. Young and colleagues reported a detailed mechanistic study with the marine sesquiterpene isonitrile 7,20-diisocyanoadociane (**84**) originally isolated from the marine sponge *Cymbastela hooperi* [87], demonstrating that it inhibited β-hematin (IC_50_ = 13nM), and thus interfered with the parasite’s heme detoxification pathway. 

As shown in Table 1 and Figure 1, potent (IC_50_ < 2 µM) to moderate (IC_50_ > 2–10 µM) *antimalarial* activity was reported for several marine natural products (**85**–**94**), although the mechanism of action for these compounds remained undertermined at the time of publication. Cheng and colleagues reported potent antiplasmodial activity in the peptide actinoramide A (**85**) isolated from *Streptomyces* species “in all five (*P. falciparum*) lines retested” [88]. Yang and colleagues discovered that a novel norditerpene diacarperoxide J (**86**) isolated from the South China Sea sponge *Diacarnus megaspinorhabdosa* was a moderate growth inhibitor of *P. falciparum* D6 clone, and observed that the peroxy functional group might be a “potential pharmacophore” [89]. Thao and colleagues characterized cembranoid diterpene laevigatol A (**87**) from several Vietnamese soft corals with potent antimalarial activity against drug sensitive *P. falciparum* strain NF54 [90]. Raju and colleagues showed that the glycohexadepsipeptide-polyketide mollemycin A (**49**) isolated from an Australian marine-derived *Streptomyces* sp. strain CMB-M0244 strongly inhibited drug sensitive *P. falciparum* strain 3D7 and multidrug resistant strain Dd2 [59]. Avilés and colleagues noted that isocyanide amphilectane-type diterpenes monamphilectines B and C (**88, 89**), isolated from the Caribbean sponge *Svenzea flava*, exhibited strong inhibitory activity against *P. falciparum* strain 3D7 [91]. Gros and colleagues observed that among the novel tricyclid alkaloids isolated from the Madagascar marine sponge *Biemna laboutei* netamine K (**90**) exhibited inhibitory activity against several *P. falciparum* strains [92]. White and colleagues reported several novel isocyano/isothiocyanate sesquiterpenes (**91–93**) isolated from the nudibranch *Phyllidia ocellata* that showed strong antiplasmodial activity against drug resistant *P. falciparum* Dd2 and 3D7 strains [93]. Chianese and colleagues assessed antimalarial activity of the novel endoperoxide polyketide (**94**) from the South China sea marine sponge *Plakortis simplex* against the *P. falciparum* chloroquine-sensitive D10 and chloroquine-resistant W2 strains [94]. 

As shown in Table 1 and Figure 1, thirteen marine compounds (**95**–**107**) isolated from bacteria, fungi, sponges, and soft corals were reported to possess bioactivity towards the so-called neglected protozoal diseases: leishmaniasis, caused by the genus *Leishmania (L.*); amebiasis, trichomoniasis, as well as African sleeping sickness (caused by *Trypanosoma (T.) brucei rhodesiense* and *T. brucei gambiense*), and American sleeping sickness or Chagas disease (caused by *T. cruzi*). 

As shown in Table 1, two reports described two *antitrypanosomal* marine chemicals (**95**, **96**) as well as their mechanisms of action. Oli and colleagues examined the mode of action of plakortide E (**95**), isolated from the sponge *Plakortis halichondrioides*, and demonstrated that it inhibited activity of *T. brucei* by a non-competitive, covalent or “mechanisms leading to slow-binding”, reversible inhibition of the parasite’s enzyme rhodesain [95]. Santos and colleagues extended the pharmacology of guanidine and pyrimidine alkaloids from the Brazilian marine sponge *Monanchora arbuscula*, and reported that batzelladine L (**96**) affected both trypomastigotes of *T. cruzi* and *L. infantum* promastigotes, demonstrating that several mechanisms including altered plasma membrane permeability, mitochondrial membrane depolarization, and increased reactive oxygen species, probably contributing to “parasite cell death” [96]. 

As shown in Table 1 and Figure 1, eleven additional marine natural products (**97**–**107**) exhibited *antileishmanial* and *antiprotozoal* activity, although their mechanisms of action remained undetermined. Abdelmohsen and colleagues reported that a new *O*-glycosylated angucycline actinosporin A (**97**), isolated from a culture of *Actinokineospora* sp. strain EG49 cultivated from a Red sea sponge *Spheciospongia vagabunda*, moderately inhibited the growth of *T. brucei brucei* [97]. Thao and colleagues isolated the terpenoid astropectenol A (**98**) from a Vietnamese marine sea star *Astropecten polyacanthus*, and observed significant activity against *T. cruzi* and *T. brucei brucei* [98]. Viegelmann and colleagues identified a new saringosterol derivative (**99**) from the Irish marine sponge *Haliclona simulans*, which demonstrated antitrypanosomal activity against *T. brucei brucei* [99]. Using genome-directed lead discovery, Schulze and colleagues contributed a novel polyene macrolactam lobosamide A (**100**) from a marine actinobacterium *Micronospora* sp. that was highly active towards the parasite *T. brucei brucei,* “likely via a parasite-specific mechanism” that remained undetermined [100]. Thao and colleagues assessed the cembranoid diterpenes lobocrasols A and C (**101, 102**), and crassumols D and E (**104, 105**), isolated from several Vietnamese soft corals, and noted that they displayed potent activity against *L. donovani* amastigotes and *T. brucei rhodesiense,* respectively [90]. Nakashima and colleagues found a new cyclopentadecane antibiotic mangromicin A (**103**) separated from the culture broth of the fungus *Lechevalieria aerocolonigenes* K10-0216 isolated from a Japanese mangrove sediment with potent activity against *T. brucei brucei* strain GUTat 3.1 [101]. Yang and colleagues characterized a new scalarane sesterterpene sesterstamide (**106**) isolated from the Paracel islands marine sponge *Hyrtios* sp. that moderately inhibited *L. donovani* promastigotes [102]. Von Salm and colleagues contributed a novel tricyclic sesquiterpenoid shagene A (**107**) from an “undescribed” soft coral collected from the “Scotia Arc in the Southern Ocean” that was moderately active against *L. donovani* [103].

Drug-resistant strains of the intracellular pathogen *Mycobacterium tuberculosis* have stimulated a search for novel drug leads with novel mechanisms of action, and, as shown in Table 1 and Figure 1, five novel marine natural products (**108**–**112**) isolated from sponges and fungi evidenced promising activity, and thus contributed to the ongoing global search for novel *antituberculosis* agents during 2014–2015.

Arai and colleagues identified a novel aaptamine class alkaloid, 2-methoxy-3-oxoaaptamine (**108**), from a marine sponge *Aaptos* sp. that demonstrated strong inhibitory activity against *M. smegmatis* in “both active growing and dormancy-inducing hypoxic conditions” [104]. Daletos and colleagues isolated cyclic peptides callyaerins A and B (**109, 110**), from the Indonesian sponge *Callyspongia aerizusa*, that demonstrated potent antibacterial activity against *M. tuberculosis*, highlighting the “potential of these compounds as promising anti-TB agents” [105]. Kumar and colleagues established that a new diarylpyrrole alkaloid denigrin C (**111**) from an extract of the Indian marine sponge *Dendrilla nigra* exhibited strong *M. tuberculosis* H_37_Rv activity “with a probable novel mechanism needed for antitubercular drug design…” [106]. Lin and colleagues characterized a racemic, prenylated polyketide dimer, oxazinin A (**112**) from a filamentous fungus isolated from the Papua New Guinea ascidian *Lissoclinum patella*, which showed activity against *M. tuberculosis* with modest activity towards human transient receptor potential channels [107].

### 2.4. Antiviral Activity

As shown in Table 1 and Figure 1, twenty-one reports were published during 2014–2015 on the *antiviral* pharmacology of marine natural products (**113**–**132**) against human enterovirus 71, human cytomegalovirus, human immunodeficiency virus type-1 (HIV-1), human herpes simplex virus (HSV), influenza virus, hepatitis B virus, murine norovirus, respiratory syncytial virus (RSV), and sindbis virus.

As shown in Table 1, five reports described antiviral marine chemicals and their mechanisms of action. González-Almela and colleagues extended the pharmacology of pateamine A (**113**), isolated from the marine sponge *Mycale* sp. by demonstrating that the compound affected the translation of genomic and subgenomic mRNAs from Sindbis virus, although “subgenomic mRNA translation (was) more resistant to pateamine A inhibition” [108]. León and colleagues identified abyssomicin 2 (**114**) from a marine-derived actinobacterium *Streptomyces* sp. that reactivated human immunodeficiency virus type-1 (HIV-1) by a protein kinase C and histone deacetylase-independent mechanism that “remains to be elucidated” [109]. Karadeniz and colleagues reported that the anti-HIV activity of the phlorotannin derivative 8,4”’-dieckol (**115**) from the Korean brown alga *Ecklonia cava* included the “ability to act against drug-resistant HIV-1 strains” by a mechanism that involved inhibition of cytopathic effects, as well as inhibition of HIV-1 reverse transcriptase enzyme [110]. Zhao and colleagues established that truncateol M (**116**) isolated from a culture of the sponge-associated fungus *Truncatella angustata* demonstrated potent activity against influenza A infections by a mechanism that targeted the virion assembly and release step, putatively becoming “a model structure of antiviral lead for further modification” [111]. Chen and colleagues determined that the alkaloid neoechinulin B (**117**) isolated from the marine-derived fungus *Eurotium rubrum* showed potent inhibition of H1N1 influenza A virus by binding to the influenza virion envelope hemagglutin, thus “disrupting its interaction with the sialic acid receptor” on host cells [112]. 

An additional 15 marine natural products (**118**–**132**), listed in Table 1 and shown in Figure 1, demonstrated antiviral activity, but the mechanism of action of these compounds remained undetermined at the time of publication. Li and colleagues isolated a novel khayanolide, thaixylomolin I (**118**) from the seeds of the Trang (South Thailand) mangrove plant *Xylocarpus moluccensis*, which inhibited potent activity against influenza virus strain H1N1 [113]. Chen and colleagues contributed a new prenylated dihydroquinolone derivative 22-*O*-(N-Me-L-valyl)-21-epi-aflaquinolone B (**119**), produced by the mycelia of an *Aspergillus* sp. fungus derived from a South China Sea gorgonian *Muricella abnormaliz,* that inhibited respiratory syncytial virus influenza A virus H1N1 “with a high therapeutic ratio” [114]. Nong and colleagues found that two novel lactones territrem D and arisugacin A (**120, 121**) from a fungus *Aspergillus terreus* SCSGAF0162 derived from a South China sea gorgonian *Echinogorgia aurantiaca* exhibited HSV-1 activity “under non-cytotoxic concentrations” [115]. Li and colleagues isolated a novel isoindolinone-type alkaloid chartarutine B (**122**) from the marine sponge-associated fungus *Stachybotrys chartarum*, which displayed moderate inhibitory activity HIV-1, noting that “side chain variation directly affected the inhibitory effects” [116]. Gupta and colleagues purified debromoaplysiatoxin (**123**) from the marine Singaporan cyanobacterium *Trichodesmium erythraeum*, which inhibited Chikungunya virus with “minimal cytotoxicity”, and probably targeted the viral replication cycle after “viral entry” [117]. Pardo-Vargas and colleagues reported that the new diterpene dolabelladienol A (**124**) isolated from the Brazilian marine brown alga *Dictyota pfaffii* had potent activity against HIV-1 and, owing to “low cytotoxicity”, appeared to be a “promising anti-HIV-1 agent” [118]. Cheng and colleagues noted that one of the dolastane diterpenes isolated from the South China sea brown alga *Dictyota plectens*, namely 13-deacetoxyamijidictyol (**125**), showed inhibitory activity against wild-type HIV-1 replication, thus proposing that “*Dictyota* algae may be a potential source of antiviral lead compounds” [119]. Yamashita and colleagues investigated the effect of two polybrominated diphenyl ethers (**22, 23**) isolated from the Indonesian marine sponge *Dysidea granulosa* on the hepatitis B virus (HBV) core promoter activity, as well as the production of HBV DNA, suggesting that they may become “candidate lead compounds for the development of anti-HBV drugs” [120]. Cao and colleagues discovered that a new steroid echrebsteroid C (**126**) from the South China sea gorgonian *Echinogorgia rebekka* evidenced high activity against respiratory syncytial virus, a common cause of lower respiratory tract disease in infants and children, as well as a high therapeutic index, thus “suggesting it might be useful as a potential antiviral agent” [121]. Jia and colleagues showed that one of two enantiomeric dimers, namely (+)-pestaloxazine A (**127**), isolated from a *Pestalotiopsis* sp. fungus derived from a soft coral, showed potent antiviral activity towards enterovirus 71, a small, single-stranded RNA virus that may cause hand, foot, and mouth disease associated with neurological complications in children and infants [122]. Eom and colleagues evaluated phlorofucofuroeckol-A (**10**), isolated from the edible brown alga *Eisenia bicyclis* against murine norovirus, a leading cause of gastroenteritis, noting that, because of its strong anti-norovirus activity and high therapeutic index, it appeared “phlorotannins could be used as a potential source of natural antiviral agents” [123]. Cheng and colleagues discovered a new *seco*-cembranoid secocrassumol (**128**) from the marine soft coral *Lobophytum crassum*, which showed significant activity against human cytomegalovirus, a common herpesvirus infection in humans [124]. Using ligand-based pharmacophore mapping, Dineshkumar and colleagues demonstrated that the polycyclic macrolide sporolide B (**129**) isolated from the marine actinomycete *Salinispora tropica* showed significant inhibition of the HIV-1 reverse transcriptase, and thus “could be a possible drug candidate for HIV” [125]. Shin and colleagues isolated two new depsipeptides stellettapeptins A and B (**130, 131**) from an extract of the Australian marine sponge *Stelletta* sp. with significant HIV-inhibitory properties, suggesting that “this class of peptides may hold promise as anti-HIV agents” [126]. Sun and colleagues characterized a new tetramic acid derivative trichobotrysin A (**132**) isolated from the culture of South China sea *Trichobotrys effuse* DFFSCS021 that inhibited herpes simplex virus type-1, responsible for lifelong oral infections in humans [127].

### 2.5. Anthelmintic Activity

As shown in Table 1, only one report was published during 2014–2015 on the anthelmintic pharmacology of marine natural products. Farrugia and colleagues isolated a 6-*N*-acyladenine alkaloid, phorioadenine A (**133**), from the southern Australian marine sponge *Phoriospongia* sp., which displayed “…nematocidal activity against *H. contortus*…slightly weaker than commercial anthelmintics levamisole and closantel”, perhaps suggesting that this compound may become a promising lead compound for the development of new anthelmintics [128].

## 3. Marine Compounds with Antidiabetic and Anti-Inflammatory Activity, and Affecting the Immune and Nervous System

Table 2 presents the 2014–2015 preclinical pharmacology of marine chemicals (**134**–**218**), which demonstrated either antidiabetic or anti-inflammatory activity, as well as affected the immune or nervous system, and whose structures are depicted in Figure 2.

### 3.1. Antidiabetic Activity

As shown in Table 2 and Figure 2, four publications reported on the mode of action of marine-derived antidiabetic compounds (**10**, **134**–**136**). Kang and colleagues contributed to the pharmacology of diabetes by noting that the marine carotenoid fucoxanthin (**134**), isolated from the marine brown alga *Ishige okamurae*, protected cells and organs from oxidative damage induced by high glucose both in vitro and in vivo, concluding that “fucoxanthin may prove to be an effective mediator to control oxidative stress in hyperglycemia” [155]. Maeda and colleagues observed that fucoxanthin and its metabolite, fucoxanthinol (**135**), improved obesity-induced inflammation in adipocyte cells with concomitant suppression of tumor necrosis factor-α and monocyte chemotactic protein-1 RNA expression, thus concluding that fucoxanthin “ameliorates glucose tolerance in the diabetic mice model” [154]. Lee and colleagues reported that octaphlorethol A (**136**) isolated from the marine brown alga *Ishige foliacea* showed a potent anti-hyperglycemic effect in mice by potently binding to α-glucosidase, an enzyme that plays a role in blood glucose control, thus demonstrating its potential use “for treatment of type 2 diabetes mellitus” [156]. You and colleagues showed that the phlorotannin phlorofucofuroeckol-A (**10**) isolated from the brown alga *Ecklonia cava* alleviated postprandial hyperglycemia in diabetic mice by a mechanism that involved significant inhibition of α-glucosidase and α-amylase, thus proposing this natural product “as a nutraceutical for diabetic individuals” [157].

An additional six marine natural products (**137**–**142**), listed in Table 2 and shown Figure 2, demonstrated antidiabetic activity, but the mechanism of action of these compounds remained undetermined at the time of publication. Safavi-Hemami and colleagues described a specialized insulin Con-Ins G1 (**137**) used for chemical warfare by the fish-hunting cone snail *Conus geographus*, which appear to have “evolved to act rapidly and potently to cause severe hypoglycemia” [158]. Yamazaki and colleagues found that the sesquiterpene dehydroeuryspongin A (**138**) isolated from the Japanese marine sponge *Euryspongia* sp. inhibited the protein tyrosine phosphatase 1B, considered a key enzyme involved in type II diabetes and obesity because it plays a role in the dephosphorylation of insulin and leptin receptors [159]. Xia and colleagues contributed a new isopimarane diterpene (**139**) isolated from the culture of the fungus *Epicoccum* sp. associated with the marine sea cucumber *Apostichopus japonicus* that potently inhibited α-glucosidase [160]. Shin and colleagues isolated a new benzothioate glycoside suncheonoside A (**140**) from a Korean marine-derived *Streptomyces* strain that promoted adiponectin production during adipogenesis in vitro, thus “suggesting antidiabetic potential” [161]. You and colleagues reported that the lumazine-containing peptide terrelumamide A (**141**), isolated from the culture broth of the Korean marine-derived fungus *Aspergillus terreus*, improved insulin sensitivity and adiponectin production in an in vitro human adipogenesis model [162]. He and colleagues characterized a polyunsaturated lipid (**142**) from the Chinese marine sponge *Xestospongia testudinaria*, which was shown to inhibit protein tyrosine phosphatase 1B, considered as a significant target for the “treatment of type II diabetes and obesity” [163].

### 3.2. Anti-Inflammatory Activity

As shown in Table 2 and Figure 2, there was a remarkable increase in anti-inflammatory pharmacology of marine compounds (**143**–**191**) during 2014–2015. The molecular mechanism of action of marine natural products (**143**–**163**) was assessed in both in vitro and in vivo preclinical pharmacological studies in twenty-two papers that used several in vitro models: the murine RAW 264.7 macrophages, a human keratinocyte cell line, a human hepatocarcinoma HepG2 cell line, primary rat brain microglia, and a murine microglia BV-2 cell line. 

Taira and colleagues evaluated the anti-inflammatory properties of alcyonolide and its congener (**143, 144**), isolated from the Okinawan soft coral *Cespitularia* sp. in lipopolysaccharide (LPS)-stimulated RAW264.7, observing inhibition of NO as well as gene expression of the proinflammatory genes inducible nitric oxide synthase (iNOS) and cyclooxygenase (COX)-2 mRNA [164]. Guo and colleagues extended the pharmacology of the terpenoid astaxanthin (**145**) by reporting that a reduction of oxidative stress in an in vivo model of rat burn injury was concomitant with a decrease in the level of malondialdehyde, an indicator of lipid peroxidation, as well as an increase of antioxidant enzymes superoxide dismutase and catalase, a “protective effect” that held “potential as a new drug treatment of severely burned patients…” [165]. Yang and colleagues reported that the polyketide 8,8′-bieckol (**146**), isolated from the edible marine brown alga *Ecklonia cava*, significantly inhibited both pro-inflammatory NO, prostaglandin E_2_ (PGE_2_), and interleukin 6 (IL-6) production, as well as gene expression by downregulating NF-κB signaling pathway and ROS accumulation in both LPS-stimulated primary macrophages and RAW 264.7 macrophages, thus demonstrating the compound’s “anti-inflammatory potential…in systemic inflammatory conditions such as sepsis” [166]. Fernandes and colleagues studied convolutamydine A (**147**), isolated from marine bryozoan *Amathia convoluta*, and two synthesized analogs, and determined that they exhibited significant in vivo and in vitro anti-inflammatory activity by a mechanism that involved reduced leukocyte migration as well as inhibition of the production of the cytokine IL-6, PGE_2_, and NO [167]. Phan and colleagues isolated a new bicyclogermacrene capgermacrene A (**148**) from the Bornean soft coral *Capnella* sp. and observed significant in vitro inhibition of NO production by RAW 264.7 macrophages by inhibition of iNOS expression, proposing this compound as a “promising iNOS inhibiting agent” [168]. Jiménez-Romero and colleagues investigated the effect of the diterpene dactyloditerpenol acetate (**149**) extracted from the Puerto Rican tropical sea hare *Aplysia dactylomela* on *E. coli* LPS-activated rat neonatal microglia in vitro, observing the potent inhibition of both thromboxane B_2_ and superoxide anion (O_2_^−^) generation, proinflammatory mediators associated with neuroinflammation, concluding that the data “support further development” of this compound [169]. Two studies extended the anti-inflammatory pharmacology of dieckol (**150**) isolated from the brown alga *Ecklonia cava*: Choi and colleagues demonstrated the compound inhibited LPS-induced iNOS expression by affecting mitogen-activated protein kinases (MAPK), “significantly p38MAPK” in the mouse macrophage 264.7 cell line in vitro [170], while Kang and colleagues demonstrated that dieckol suppressed production of macrophage-derived chemokine, C–C motif chemokine 22, an inflammatory chemokine that controls leukocyte movements by down-regulating the activation of the signal transducer and activator of transcription (STAT)1 signaling pathway in human keratinocytes [171]. Lin and colleagues characterized the anti-inflammatory effects of the diterpene excavatolide B (**151**) isolated from the cultured Formosan marine gorgonian *Briareum excavatum* and observed that, in vitro, it inhibited iNOS and COX-2 mRNA expression in LPS-treated murine RAW 264.7 macrophages, while in vivo it attenuated carrageenan-induced rat paw inflammation and pain, thus concluding that “excavatolide B may serve as a useful therapeutic agent for the treatment of acute inflammation” [172]. Chen and colleagues investigated the antinociceptive properties of flexibilide (**152**), isolated from the Australian soft coral *Sinularia flexibilis* in the rat chronic injury model of neuropathic pain, observing significant analgesic effects concomitant with suppression of iNOS expression in microglia and astrocytes in the spinal dorsal horn, accompanied with upregulation of transforming growth factor-β1 (TGF-β1), “suggesting involvement of TGF-β1 in the anti-neuroinflammatory and analgesic effects” [173]. Kim and colleagues reported that a polyhydroxyflavone (**153**) isolated from the marine alga *Hizikia fusiforme* suppressed LPS-stimulated RAW 264.7 cells’ release of pro-inflammatory cytokines, as well as both iNOS and COX-2 expression, by attenuating nuclear transcription factor-κB (NF-κB) translocation, and thus might become a “potential therapeutic agent for patients with, or at risk of, septic shock or other inflammatory diseases” [174]. Wijesinghe and colleagues evaluated 5β-hydroxypalisadin B (**154**), a brominated secondary metabolite isolated from the Malaysian marine red alga *Laurencia snackeyi*, on LPS-stimulated RAW 264.7 macrophages and observed significant reduction of several pro-inflammatory cytokines, NO, and PGE_2_ generation, and thus concluded that the compound might help development of “an active ingredient in pharmaceutical, nutraceutical…” applications [175]. Huang and colleagues characterized two novel biscembranes glaucumolides A and B (**155, 156**) from the cultured soft coral *Sarcophyton glaucum* that significantly inhibited O_2_^-^ generation and elastase release in human neutrophils, while also reducing expression of iNOS and COX-2 in LPS-treated murine RAW 264.7 macrophages, concluding that these two compounds “might be useful for future biomedical applications” [176]. Yu and colleagues determined that the effects of phlorofucofuroeckol-B (**157**), isolated from the marine alga *Ecklonia stolonifera*, on the decreased production of pro-inflammatory mediators by LPS-stimulated BV-2 microglia cells, as well as reduced COX-2 and iNOS expression, resulted from inhibition of the iκB-α/NF-κB and Akt/ERK/JNK pathways, thus proposing that this compound might be “considered as a therapeutic agent against neuroinflammation” [177]. Babskota and colleagues isolated a new phosphatidylglycerol (**158**) from an extract of the marine red alga *Palmaria palmata*, also commonly known as dulse, which strongly inhibited NO release from LPS treated murine RAW 264.7 macrophages, probably by a mechanism that down-regulated iNOS, thus suggesting that “consumption of dulse as a functional food may help to reduce inflammation associated with various diseases“ [178]. Itoh and coleagues showed that reduced scytonemin (**159**) isolated from the cosmopolitan colonial cyanobacterium *Nostoc commune* strongly inhibited LPS and interferon-γ-induced NO production in murine macrophage RAW 264.7 macrophages, by generating reactive oxygen species by activation of the phosphatidylinositol-3-kinase/Akt and the p38 mitogen-activated protein kinase/nuclear factor erythroid 2-related factor 2 signaling pathways [179]. Lillsunde and colleagues reported that a norcembranoid sinuleptolide (**160**) isolated from the Indian soft coral *Sinularia kavarattiensis* potently modulated both morphology and release of pro-inflammatory and anti-inflammatory mediators by LPS-treated rat primary microglial cells in vitro, thus decreasing microglia activation, which has been hypothesized to be involved in the “progression of chronic neurodegenerative diseases.. and central nervous system (CNS) homeostasis” [180]. Thao and colleagues contributed a new polyhydroxylated steroid sarcopanol A (**161**) from the Vietnamese soft coral *Sarcophyton pauciplicatum* that inhibited tumor necrosis factor (TNF)-α and interferon (IFN)γ-induced expression of COX-2, iNOS, and intercellular adhesion molecule-1 (ICAM-1) in the spontaneously transformed immortal human keratinocyte cell line HaCaT via inhibition of NF-κB signaling pathway activation [181]. Thao and colleagues investigated the diterpenoid sinumaximol (**162**), isolated from the marine soft coral *Sinularia maxima*, and determined that it significantly inhibited TNF-α-induced NF-κB transcriptional activity in a human hepatocarcinoma HepG2 cell line, while concomitantly inhibiting the expression of pro-inflammatory iNOS and ICAM-1mRNA expression, thus supporting the “therapeutic potential as anti-inflammatory” of this compound [182]. Quang con colleagues determined that tanzawaic acid A (**163**), isolated from a marine fungus *Penicillium* sp. SF-6013 derived from the Pacific sea urchin *Brisaster latifrons*, inhibited both NO and PGE_2_ production from LPS-activated murine BV-2 microglia cells and RAW 264.7 murine macrophages, while suppressing iNOS and COX-2 expression and inhibiting protein tyrosine phosphatase 1B [183]. 

In contrast to the marine compounds (**143**–**163**) with described anti-inflammatory mechanisms of action described in the preceding paragraph, and as shown in Table 2, for marine compounds (**164**–**191**), only anti-inflammatory activity (IC_50_) was reported, but the molecular mechanism of action of these marine-derived compounds remained undetermined at the time of publication: a new highly oxygenated meroterpene aspertetranone D (**164**) isolated from the marine algal-associated fungus *Aspergillus* sp. ZL0-1b14 [184]; a novel 6,12-dichlorobriarene diterpenoid briarenolide J (**165**), two briarane diterpenoids briarenolides K and L (**166, 167**), and the briarenolides U–W (**168**–**169**), all compounds isolated from the Taiwanese octocoral *Briareum* sp. [185,186,187]; two new briarane diterpenoids briaviolides E and I (**170, 171**), isolated from the Taiwanese soft coral *Briareum violacea* [188]; a new pigmented phenazine compound dermacozine H (**172**), isolated from the actinomycete *Dermacoccus abyssi* sp. nov. strain MT1.1, isolated from a Mariana Trench sediment at a depth of 10,898 m [189]; a novel sesquiterpene dysifragilone A (**173**), isolated from the South China Sea sponge *Dysidea fragilis* [190]; a new xenicane 4α-hydroxypachylactone (**174**), isolated from a Chinese brown alga *Dictyota plectens* [191]; a polyketide comaparvin (**175**), isolated from the Taiwanese marine crinoid *Comanthus bennetti* [192]; two novel eunicellin-type diterpenoids hirsutalins N and S (**176, 177**) and a new tocopherol-derived metabolite hirsutocospiro A (**178**), isolated from the Taiwanese soft coral *Cladiella hirsuta* [193,194,195]; a new cembranoid isosinulaflexiolide K (**179**), isolated from cultured Taiwanese soft corals *Sinularia sandensis* and *Sinularia flexibilis* [196]; a novel 9,11-secogorgosteroid klyflaccisteroid F (**180**), isolated from the Taiwaneses soft coral *Klyxum flaccidum* [197]; new eunicellin-type diterpenoids krempfielins N (**181**), Q (**182**), and R (**183**), isolated from a Taiwanese soft coral *Cladiella krempfi* [198,199]; a methylfarnesylquinone (**184**), isolated from the Taiwanese marine brown alga *Homeostrichus formosana* [200]; a novel steroid monanchosterol B (**185**), isolated from the Korean sponge *Monanchora* sp. [201]; a oxygenated steroid derivative (**186**), isolated from the Vietnamese starfish *Protoreaster nodosus* [202]; a new spirobisnaphthalene rhytidenone C (**187**), isolated from an extract of a cultured fungal endophyte *Rhytidhysteron* sp. AS21B isolated from a Thailandese mangrove area [203]; a known terpenoid sarcocrassocolide E (**188**), isolated from a Taiwanese soft coral *Sarcophyton crassocaule* [204]; a new cembrane diterpenoid sinulacembranolide A (**189**), isolated from the Taiwanese soft coral *Sinularia gaweli* [205]; a new eudesmane-type sesquiterpene thomimarine B (**190**), isolated from the fungus *Penicillium thomii* KMM 4667 isolated from the Japanese sea grass *Zostera marina* [206]; and a novel diterpenoid tortuosene A (**191**), isolated from the Taiwanese soft coral *Sarcophyton tortuosum* [207].

### 3.3. Marine Compounds with Activity on the Immune System

As shown in Table 2 and Figure 2, the preclinical pharmacology of marine compounds that affected the *immune* system showed a decline, as previously reported in this series.

Kwan and colleagues reported that the peptide grassypeptolide A (**192**), isolated from the marine cyanobacterium *Lyngbya confervoides,* inhibited IL-2 production and proliferation of activated T cells by inhibiting the protease dipeptidyl peptidase 8, probably by binding at inner cavity of the enzyme at two distinct sites [208]. Wang and colleagues isolated a pair of novel bisheterocyclic quinolone-imidazole alkaloids (+)- and (^_^) spiroreticulatine (**193**) from the South China sea sponge *Fascaplysinopsis reticulata*, which showed inhibition of IL-2 production by Jurkat T cells [209]. Kicha and colleagues determined that the cyclic steroid glycoside luzonicoside A (**194**), isolated from the starfish *Echinaster luzonicus*, potently enhanced lysosomal activity, ROS level elevation, and NO synthesis in RAW 264.7 murine macrophages, thus seeming “promising for further investigation as a potent immunomodulatory agent” [210]. Pislyagin and colleagues investigated a triterpene glycoside typicoside C_1_ (**195**), isolated from the sea cucumber *Actinocucumis typica*, and observed that it demonstrated strong immunostimulatory effect on ROS formation in mouse peritoneal macrophages in vitro, with concomitant low cytotoxicity [211]. 

### 3.4. Marine Compounds Affecting the Nervous System

As shown in Table 2 and Figure 2, in 2014–2015, the preclinical marine *nervous* system pharmacology of compounds (**196**–**211**) described several mechanism of action: at the nicotinic acetylcholine receptor and potassium channels, with conopeptides, and in models of antinociception and neuroprotection. 

Four marine compounds were shown to bind nicotinic acetylcholine receptors (nACHR) (**199**, **207**, **210**) and potassium (K^+^) channels (**208**). Kasheverov and colleagues determined the effect of 6-bromohypaphorine (6-BHP) (**199**), isolated from the marine nudibranch mollusk *Hermissenda crassicornis*, on different nAChR, demonstrating that, because 6-BHP competed with α-bungarotoxin for binding to the human α7 nAChR, it was the “first low-molecular weight compound from (a) marine source which (was) an agonist of the nACHR subtype” [215]. Bourne and colleagues conducted detailed studies to determine the molecular pharmacology of the macrocylic imine phycotoxin pinnatoxin A (**207**), originally isolated from the digestive glands of the mollusk *Pinna attenuata*, towards neuronal α7nACHR, observing that the bicyclic EF-ketal ring was a novel binding determinant for mediating polar versus non-polar interactions, and thus is able to “confer nAChR subtype selectivity… (of) these prevalent marine biotoxins” [223]. Rodríguez and colleagues discovered a novel peptide PhcrTx1 (**208**) from the sea anemone *Phymanthus crucifer* that inhibited voltage-gated K^+^ ion channels, including acid-sensing ion channel (ASIC) (IC_50_ = 100 nM), and that would represent “the first member of a new structural group of sea anemone toxins acting on ASIC” [224]. Aráoz and colleagues extended the pharmacology of the “fast-acting” lipophilic marine toxin 13,19-desmethyl spirolide C (**210**), extracted from cultures of the dinoflagellate *Alexandrium ostenfeldii*, defining the mode of action and molecular targets using in vitro electrophysiological experiments, and thus showing that the toxin blocked human neuronal nACHR (IC_50_ = 0.2 nM) with high affinity, observations supported by molecular docking experiments “highlighting the nicotinic basis of the neurotoxicity of (this toxin) to mammal(ian)….peripheral and central nervous system” [227].

Three studies extended the pharmacology of conopeptides (**201**–**203**). Wang and colleagues discovered a novel α-conotoxin Mr1.7 (**201**) in the venom of the marine snail *Conus marmoreus* that inhibited α3β2, α9α10, and α6/α3β2β3 nACHR subtypes (IC_50_ = 53.1, 185.7, and 284.2 nM, respectively), noting that the PE residues at the N-terminal sequence of Mr1.7 were “important for modulating activity and selectivity” [217]. Zhou and colleagues reported the expression and sodium channel activity of peptide It16a (**202**), a novel framework XVI conotoxin from the M-superfamily isolated from the worm-hunting snail *C. litteratus,* and, using a variety of electrophysiological techniques, demonstrated that it preferentially inhibited voltage-gated Na^+^ channels (apparent IC_50_ = 1 µM) in mammalian sensory neurons, with the authors noting “It16a …has similar function as µ-conotoxins” [218]. Li and colleagues extended the pharmacology of Vt3.1 conotoxin (**203**), isolated from the venom of the marine cone snail *C. vitulinus*, and demonstrated that it preferentially inhibited large conductance, voltage, and Ca^2+^ activated K^+^ (BK) channels containing the *β*4 subunit (IC_5_ = 8.5 µM), which appears to be present in brain and neuronal functions by a mechanism that required electrostatic interactions with the channel protein, making it an excellent tool “uniquely suited in neuroscience involving BK channels” [219]. 

Two studies reported marine compounds (**198**, **200**) that contributed to nociceptive pharmacology. Cavalcante-Silva and colleagues assessed the mechanism involved in in vivo antinociception produced by the bisindole alkaloid caulerpine (**198**), isolated from the marine alga *Caulerpa*, demonstrating that, in the in vivo murine writhing test, the effect was likely mediated by “pathways involving α2-adrenoceptors and 5-HT3 receptors”, thus proposing cualerpine as a possible “dual-action analgesic drug(s)” [214]. Chen and colleagues investigated the anti-neuropathic properties of the antimicrobial peptide piscidin (**200**) and observed that the compound demonstrated in vivo anti-nociceptive effects in a rat model of neuropathy by a signaling mechanism that suppressed up-regulation of interleukin-1 in microglia and phosphorylated mammalian target of rapamycin in astrocytes, concluding it “may have potential for development as an alternative pain-alleviating agent” [216].

The neuroprotective activity of marine compounds (**196**, **197**, **205**, **206**, **211**) was reported in five studies. Wu and colleagues observed that the novel (Z)-7,4′-dimethoxy-6-hydroxy-aurone-4-O-*β*-glucopyranoside (**196**) isolated from the endophytic fungus *Penicillium citrinum* of the mangrove *Bruguiera gymnorrhyza* derivative decreased 1-methyl-4-phenylpyridium-induced neurotoxicity in rat pheochromocytoma PC12 cells in vitro by a mechanism that elevated mitochondrial membrane potential, decreased DNA fragmentation, and inhibited apoptosis [212]. Hjornevik and colleagues completed an extensive in vitro “neurotoxicological” study with the marine algal toxin azaspiracid-1 (**197**) and observed rat PC12 cells’ differentiation-related morphological changes associated with the expression of the PC12-associated neuronal differentiation marker peripherin on neurite-like processes, suggesting this molecule “triggers a differentiation process” [213]. Yamagishi and colleagues explored the structure–activity relationship of LLG-3 (**205**), a ganglioside isolated from the starfish *Linchia laevigata*, and discovered that the methyl group at C8 of the terminal sialic acid residue was of critical significance for neuritogenic activity. Furthermore, detailed signaling studies revealed the “activation of mitogen-activated protein kinase signaling pathway” [221]. Cassiano and colleagues, using chemical proteomics, noted that the terpenoid heteronemin (**206**), isolated from the marine sponge *Hyrtios* sp., targeted TDP-43, a major component of inclusions that characterize amyotrophic lateral sclerosis and front-temporal lobar degeneration, by lowering its affinity “towards nucleic acids”, and thus becoming a “relevant chemical tool in the study of TDP-43 related processes” [222]. Shimizu and colleagues provided the “first report” that the pro-electrophilic sesquiterpene zonarol (**211**), isolated from the Japanese brown alga *Dictyopteris undulata*, provided neuroprotection by activating the nuclear factor (erythroid-derived-2)-like 2/antioxidant responsive element Nrf2/ARE pathway, inducing phase-2 enzymes and providing oxidative stress protection to cerebrocortical neurons in vitro, concluding that the compound “represents a lead compound for the treatment of chronic neurodegenerative diseases associated with oxidative stress” [228].

As shown in Table 2, three marine compounds were shown to modulate other molecular targets, that is, γ-aminobutyric acid (GABA) receptor (**209**), and the acetylcholinesterase (**204**) and butyrylcholinesterase enzyme (**10**). Lee and colleagues discovered that the pigment echinochrome A (**204**), isolated from the sea urchin *Scaphechinus,* inhibited acetylcholinesterase (IC_50_ = 16.4 µM) by an irreversible and uncompetitive mechanism that might be useful in “treating acetylcholine-limited diseases”, such as Alzheimer’s disease and “other forms of dementia” [220]. Eltahawy and colleagues isolated of a new ceramide (**209**) from the Red sea soft coral *Sarcophyton auritum,* which demonstrated antiepileptic activity in vivo with a central nervous system depressing mechanism that appeared to involve “GABA receptor modulation rather than serotonin receptor inhibition” [226]. Choi and colleagues reported that the polyphenol phlorofucofuroeckol-A (**10**), isolated from the Korean brown alga *Ecklonia cava*, potently inhibited butyrylcholinesterase, a novel target for Alzheimer’s disease, suggesting that “phlorotannins … to be very promising medicinal compounds” [225].

In contrast to the marine compounds affecting the nervous system with investigated mechanisms of action discussed above, and as shown in Table 2, for marine compounds (**120**, **121**, **212**-**218**), only an IC_50_ was reported, but the molecular mechanism of action of these compounds remained undetermined at the time of publication: an Australian marine-sponge *Aplysinella* sp.-derived aplysinellamide-1 (**212**) [229], the novel lactones territrem D, and arisugacin A (**120, 121**) from a fungus *Aspergillus terreus* SCSGAF0162 derived from a South China sea gorgonian *Echinogorgia aurantiaca* [115]; an Indian marine cone snail *Conus araneosus* ar3j peptide (**213**) [230]; a new steroid (**214**) from a fungus *Dichotomomyces cejpii* isolated from an Australian marine sponge *Callyspongia* cf. C. *flammea* [231]; a novel genuanine (**215**) isolated from Cape Verde marine cone snail *Conus genuanus* [232]; a bromotyrosine alkaloid homoaerothionin (**216**) isolated from the Thai sponge *Acanthodendrilla* sp. [233]; a new alkyl amide mooreamide A (**217**) from the Papua New Guinean marine cyanobacterium *Moorea bouillonii* [234]; and a hydroxyoctaprenyl 1′,4′-hydroquinone (**218**) isolated from the Italian marine sponge *Sarcotragus spinosulus* [235].

## 4. Marine Compounds with Miscellaneous Mechanisms of Action

The 2014–2015 preclinical pharmacology of 83 marine compounds (**219**–**300**) with miscellaneous mechanisms of action is shown in Table 3, with their corresponding structures presented in Figure 3. Because, at the time of publication, a comprehensive pharmacological characterization of these compounds remained unavailable, their assignment to a particular drug class will probably require further investigation.

As reported in the peer-reviewed literature, Table 3 presents the pharmacological activity, an IC_50_, and a molecular mechanism of action of the following marine natural compounds: sea anemone *Aiptasia diaphana* toxic peptide AdE-1 (**219**) [236]; sponge alkaloid aaptamine (**220**) [237]; dinoflagellate *Amphidinium* sp. polyketide amphirionin-4 (**221**) [238]; algal terpenoid astaxanthin (**145**) [239]; sponge alkaloids bastadins 6 and 16 (**222, 223**) [240]; brown alga *Eisenia bicyclis* polyketide 6,6-bieckol (**224**) [241]; *Streptomyces* sp. strain CNH-287 alkaloid (-)-chlorizidine A (**225**) [242]; soft coral *Cladiella australis* dihydroaustrasulfone alcohol (**226**) [243,244]; edible brown alga *Ishige okamurae* terpenoid diphlorethohydroxycarmalol (**227**) [245]; sea urchin *Scaphechinus mirabilis* alkaloid echinochrome A (**204**) [246,247]; brown alga *Ecklonia stolonifera* polyketide eckol (**228**) [248]; sponge derived fungus *Dichotomomyces cejpii* terpenoid emindole SB (**229**) [249]; Arctic *Streptomyces nitrosporeus* YBH10-5 farnesylquinone (**230**) [250]; fungus *Stachybotrys longispora* FG216 pyrano indolone alkaloid fibrinolytic compound 1 (**231**) [251]; fungus *Paecilomyces formosus* formosusin A (**232**) [252]; brown alga *Ecklonia cava* phlorotannin fucodiphlorethol G (**233**) [253]; green alga *Spirogyra* sp. polyphenol gallic acid (**234**) [254]; cyanobacterium *Schizothrix* sp. gallinamide A (**235**) [255]; sponge *Stylissa aff. carteri* girolline (**236**) [256]; sponge *Spongionella* sp. terpenoids gracilins H, A, and L (**237–239**) [257]; new anemone *Heteractis crispa* Kunitz-type polypeptides HCRG1 and HCRG2 (**240, 241**) [258]; new sponge *Hyrtios* sp. sesterterpenoid 2 (**242**) [259]; sponge *Ircinia ramose* irciniastatin A (**243**) [260]; ascidian *Eudistoma* cf. *rigida* polyketide lejimalide C (**244**) [261]; red alga *Laurencia brongniartii* brominated indole (**245**) [262]; sponge *Neopetrosia* sp. pyridine nucleoside neopetroside A (**246**) [263]; edible brown alga *Ecklonia stolonifera* phlorotannin phlorofucofuroeckol-A (**10**) [264]; sponge *Theonella swinhoei* peptide polytheonamide B (**247**) [265]; green alga *Codium fragile* terpenoid siphonaxanthin (**248**) [266]; deep-sea derived fungus *Spiromastix* sp. MCCC 3A00308 spiromastixones J and L (**57, 249**) [267]; bacteria *Thalassospira* sp. CNJ328 and *Tistrella bauzanensis* TIO7329 thalassospiramide C (**250**) [268]; mangrove fungus *Xylaria* sp. xyloketal B (**251**) [269,270]; and sponge *Xestospongia testudinaria* brominated polyunsaturated lipid (**252**) [271].

Also consolidated in Table 3 is the pharmacological activity (IC_50_ for enzyme or receptor inhibition) of marine-derived compounds (**253**–**300**), but the mechanism of action remained undetermined at the time of publication: a dinoflagellate *Dinophysis acuminate* new polyether macrolide acuminolide A (**253**) [272]; fungus *Alternaria alternata* alternariol derivatives (**255**–**257**) [136]; bacterium *Streptomyces* sp. linear peptide ahpatinin Ac (**254**) [273]; ascidian *Aplidium* sp. new dipeptide apliamide D (**258**) [274]; fungi *Penicillium thomii* and *P. lividum* new meroterpenoids austalides 4 and 9 (**259, 260**) [275]; *Streptomyces axinellae* axinelline A (**261**) [276]; dinoflagellate *Karenia brevisulcata* polyether brevisulcatic acid-4 (**262**) [277]; green alga *Caulerpa racemose* 4′,5′–dehydrodiodictyonema A (**263**) [278]; sponge *Dactylospongia metachromia* sesquiterpenes nakijiquinone N (**264**), nakijinol C (**266**), and known analog 18-hydroxy-5-*epi*-hyrtiophenol (**265**) [279]; ascidian *Didemnum* sp. spiroketals didemnaketals D and E (**267, 268**) [280]; sponge *Dysidea avara* sesquiterpene dysiquinol D (**269**) [281]; brown alga *Laminaria japo*nica terpenoid fucoxanthin (**134**) [282]; sponge *Xestospongia testudinaria* halenaquinol sulfate (**270**) [283]; sponge *Xetospongia* sp. halenaquinone derivative, 1-hydroxyethylhalenaquinone (**271**) [284]; sponge *Stylissa massa* and *S. flabelliformis* bromopyrrole alkaloids (**272**–**274**) [285]; sponge *Callyspongia* sp. hymenialdisine (**275**) [286]; sponge *Hyattella* sp. hyattellactone A (**276**) [287]; sponge *Hippospongia lachne* sesterterpene hippolide derivative (**277**) [288]; sponge *Suberea ianthelliformis* ianthelliformisamines A, B, and C (**278, 7, 8**) [289]; sponge *Plakortis* cfr. *lita* sterols incisterols A5 and A6 (**279, 280**) [290]; soft coral *Lobophytum crissum* cembranoid diterpene 2,16:7*S*,8*S*,-diepoxy 1,3,11,15-cembratetraene (**281**) [291]; red alga *Laurencia okamurai* laurene-type sesquiterpenoid laurokamurane A (**282**) [292]; gorgonian *Echinogorgia pseudossapo* alkaloid malonganenone L (**283**) [293]; fungus *Hansfordia sinuosae* sesquiterpenoid punctaporonin K (**284**) [294]; fungus *Aspergillus versicolor* ZLN-60 cyclic peptide psychrophilin G (**285**) [295]; green alga *Caulerpa racemosa* bisindole alklaloid racemosin C (**286**) [296]; soft coral *Sarcophyton ehrenbergi* prostaglandin derivative sarcoehrendin B (**287**) [297]; soft coral *Sarcophyton trocheliophorum* Marenzeller diterpene sarsolilide A (**288**) [298]; fungus *Stachybotry* sp. HH1 ZSDS1F1-2 xanthone derivative stachybogrisephenone B (**289**) [299]; brown alga *Sargassum thunbergii* alkapolyene 1 (**290**) [300]; red alga *Palmaria palmata* mycosporine-like amino acid shinorine (**291**) [301]; soft coral *Sinularia* sp. cyclopentenone sinularone D (**292**) [302]; fungus *Stachybotrys chartarum N*-(2-benzenepropanoic acid) stachybotrylactam (**293**) [303]; sponge *Petrosia corticata* meroditerpenoids strongylophorine -13/-14 (**294**) [304]; sponge *Theonella swinhoei* steroid swinhoeisterol A (**295**) [305]; brown alga *Sargassum thunbergii* thunberol (**296**) [306]; sponge *Xestospongia vansoesti* meroterpenoid xestosaprol O (**297**) [307]; sponge *Monanchora pulchra* bisguanidine alkaloid urupocidin A (**298**) [308]; sponge *Luffariella variabilis* β-carboline alkaloid variabine B (**299**) [309]; and cyanobacterium *Leptolyngbya* sp. yoshinone A (**300**) [310].

## 5. Reviews on Marine Pharmacology and Pharmaceuticals

In 2014–2015, several reviews covered general and/or specific areas of marine preclinical pharmacology: (a) ***Marine pharmacology and marine pharmaceuticals:*** new marine natural products and relevant biological activities published in 2014 and 2015 [311,312]; marine peptides, bioactivities and applications [313]; bioactive terpenes from marine-derived fungi [314]; bioactive marine natural products from actinobacteria with unique chemical structures [315]; Baltic cyanobacteria as a source of biologically active compounds [316]; biological targets of marine cyanobacteria natural products [317]; marine mussels as a source for bioactive compounds for human health [318]; pharmacological potential of cephalopod ink in drug discovery [319]; pharmacologically active Brazilian octocorals [320]; bioactive natural products isolated from marine microorganisms from Brazil [321]; statistical analysis of marine natural product bioactivity from 1985–2012 [322]; metagenomics and marine natural products drug discovery [323]; new horizons for selected marine natural products as drug leads [324]; marine-sourced agents in clinical and late preclinical development [325]; the global marine pharmaceutical pipeline in 2019: approved compounds and those in Phase I, II, and III of clinical development https://www.midwestern.edu/departments/marinepharmacology.xml. (b) ***Antimicrobial marine pharmacology****:* biophysical properties of anti-lipopolysaccharide antimicrobial peptides isolated from marine fish [326]; marine peptides and their anti-microbial activities [327]; marine membrane-active peptides as antimicrobials [328]; marine fungi antibacterial compounds [329]. (c) ***Antiviral marine pharmacology***: marine natural products with antiviral potential [330]; antiviral activity in marine fungi-derived natural products [331]. (d) ***Antiprotozoal and antimalarial marine pharmacology****:* antiprotozoal activity in marine natural products isolated from marine algae [332]; marine indole alkaloids as potential leads for antiprotozoal drugs [333]; antimalarial potency of the manzamine β-carboline alkaloids [334]. (e) ***Immuno- and anti-inflammatory marine pharmacology****:* marine diterpenoids as potential anti-inflammatory agents [335]; microalgae bioactive compounds for inflammation and cancer [336]. (f) ***Cardiovascular and antidiabetic marine pharmacology***: marine-derived natural products as a source of cardiovascular protective agents [337]; antioxidant phlorotannins derived from marine algae [338]; antioxidant carotenoids isolated from marine Gram-positive bacteria [339]; brown alga-derived fucoxanthin for diabetes therapy [340]; bioactive compounds from seaweed for diabetes [341]. (g) ***Nervous system marine pharmacology****:* astaxanthin as a potential neuroprotective agent [342]; origin, distribution, toxicity, and therapeutic uses of the marine neurotoxin tetrodotoxin [343]; marine natural products with neuroprotective activity [344]; marine-terpenoid gracilins as promising compounds for Alzheimer’s disease [345]; new marine drugs for Alzheimer’s disease treatment [346]. (h) ***Miscellaneous molecular targets and uses***: matrix metalloproteinase inhibitors isolated from edible marine algae [347]; marine natural products that targeting apoptosis signaling pathways [348]; scytonemin and emerging biomedical applications [349]; antiobesity effects of the carotenoid fucoxanthin [350]; therapeutic potential of astaxanthin [351]; pharmacological properties of marine coumarins [352].

## 6. Conclusions

The current marine pharmacology 2014–2015 review is a sequel to the marine *preclinical* pharmacology pipeline review series initiated in 1998 [1,2,3,4,5,6,7,8,9], and consolidates the peer-reviewed preclinical marine pharmacological literature published during 2014–2015. The global preclinical marine pharmacology research involved chemists and pharmacologists from 43 countries, namely, Australia, Austria, Bangladesh, Belgium, Brazil, Canada, China, Colombia, Costa Rica, Cuba, Denmark, Egypt, Finland, France, French Polynesia, Germany, Hungary, India, Indonesia, Ireland, Israel, Italy, Japan, Malaysia, Mexico, the Netherlands, New Zealand, Norway, Papua New Guinea, Portugal, Russian Federation, Saudi Arabia, Singapore, South Africa, South Korea, Spain, Sri Lanka, Switzerland, Taiwan, Thailand, United Kingdom, Vietnam, and the United States. Thus, during 2014–2015, the marine *preclinical* pharmaceutical pipeline continued to provide novel pharmacology that provided novel leads for the marine *clinical* pharmaceutical pipeline. As shown at the global marine pharmaceutical pipeline website, https://www.midwestern.edu/departments/marinepharmacology.xml, there are currently 9 approved marine-derived pharmaceuticals, and an additional 31 compounds are either in Phase I, II, and III of *clinical* pharmaceutical development. 

## Figures and Tables

**Figure 1 marinedrugs-18-00005-f001:**
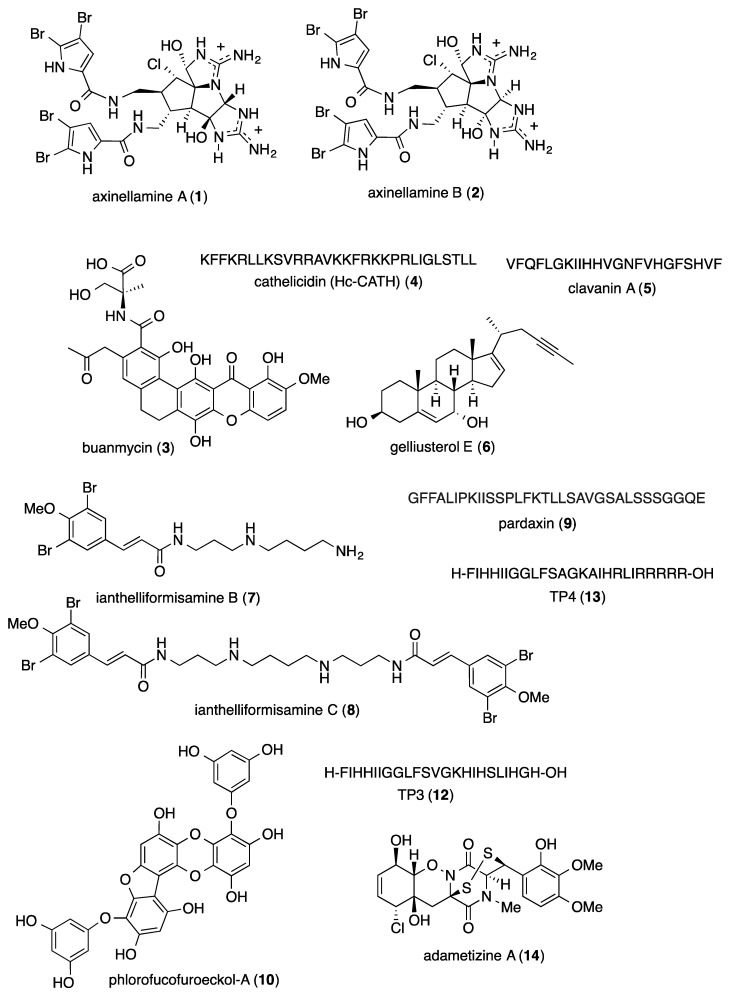
Marine pharmacology in 2014–2015: marine compounds with antibacterial, antifungal, antiprotozoal, antituberculosis, antiviral, and anthelmintic activities.

**Figure 2 marinedrugs-18-00005-f002:**
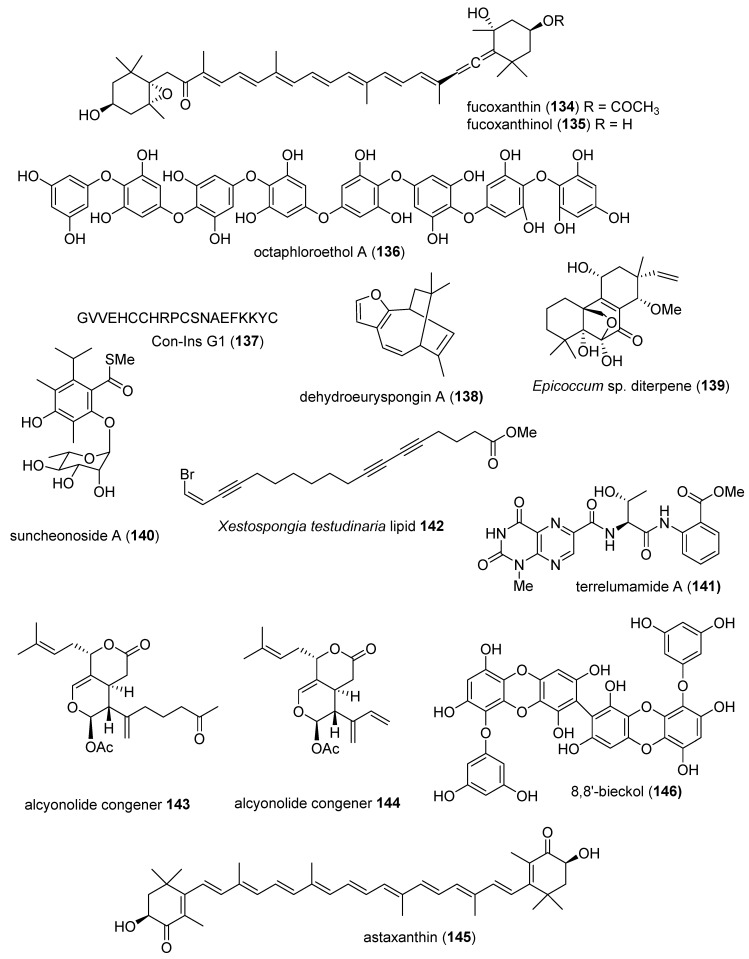
Marine pharmacology in 2014–2015: marine compounds with antidiabetic and anti-inflammatory activity, and affecting the immune and nervous system.

**Figure 3 marinedrugs-18-00005-f003:**
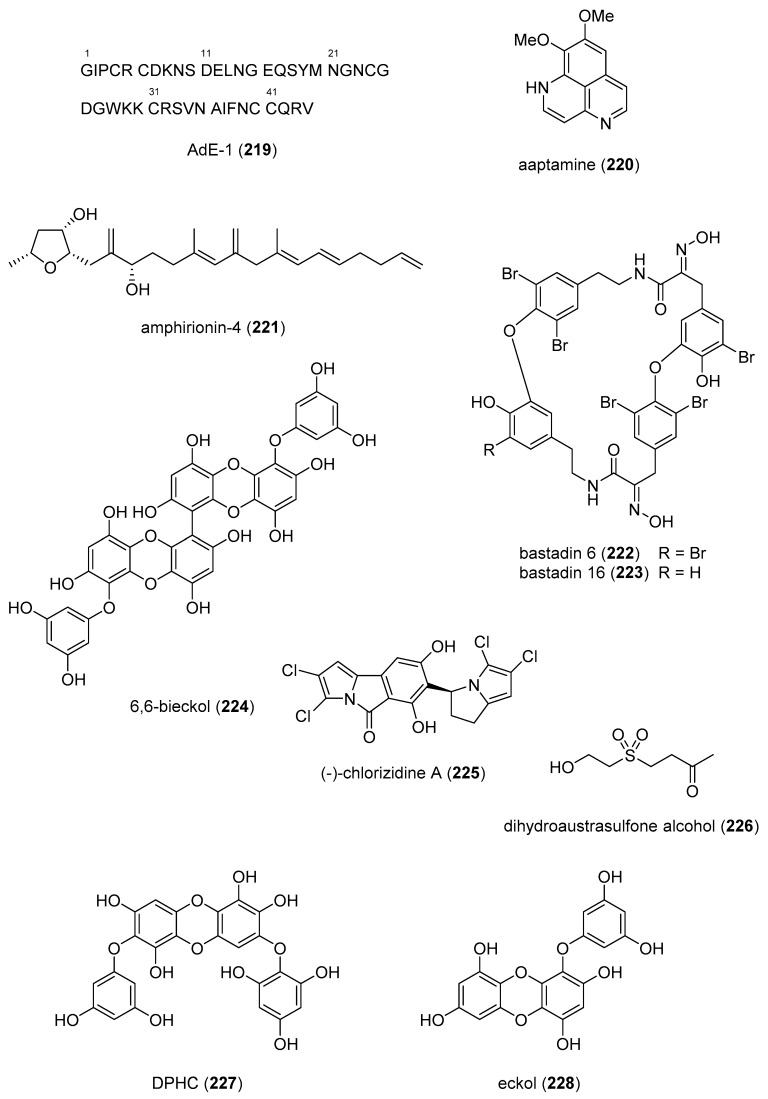
Marine pharmacology in 2014–2015: marine compounds with miscellaneous mechanisms of action.

**Table 1 marinedrugs-18-00005-t001:** Marine pharmacology in 2014–2015: marine compounds with antibacterial, antifungal, antituberculosis, antiprotozoal, antiviral, and anthelmintic activities.

Drug Class	Compound/Organism ^a^	Chemistry	Pharmacologic Activity	IC_50_ ^b^	MMOA ^b^	Country ^c^	References
Antibacterial	axinellamines A and B (**1**, **2**)/sponge	Alkaloid ^f^	Gram-positive and negative inhibition	0.5–32 μg/mL ^+^	Normal cellular division inhibition	USA	[24]
Antibacterial	buanmycin (**3**)/bacterium	Polyketide ^d^	*S. enterica* inhibition	0.7 μM ^+^	Sortase A inhibition	S. KOR	[25]
Antibacterial	cathelicidin (**4**)/sea snake	Peptide ^f^	Gram-positive and negative inhibition	0.16–20.7 μg/mL ^+^	Membrane morphology alteration	CHN	[26]
Antibacterial	clavanin A (**5**)/ascidian	Peptide ^f^	*S. aureus* and *E. coli* infection inhibition	10 mg/kg ***	IL-6 and TNF-α inhibition	BRA	[27]
Antibacterial	gelliusterol E (**6**)/sponge	Terpenoid ^e^	*C. trachomatis* inhibition	2.34 μM	OmpA protein inhibition	EGY, GBR	[28]
Antibacterial	ianthelliformisamimes B and C (**7**, **8**)/sponge	Alkaloid ^f^	Enhanced antibiotics against *E. aerogenes, P. aeruginosa, K. Pneumoniae* MDR strains in vitro	3.12–12.5 μM *	Enhancement of drug transporters	FRA	[29]
Antibacterial	pardaxin (**9**)/flatfish	Peptide ^f^	*MR S. aureus* inhibition *in vivo*	8 mg/mL *	MCP-1, IL-6, and TNF-α inhibition	TWN	[30]
Antibacterial	phlorofucofuroeckol-A (**10**)/alga	Polyketide ^d^	*MR S. aureus* inhibition	32 μg/mL ^+^	PBP2a suppresion	S. KOR	[31]
Antibacterial	salinamide F (**11**)/bacterium	Peptide ^f^	*E. coli* inhibition	0.2 μg/mL ^+^	RNAP inhibition	USA	[32]
Antibacterial	piscidins (**12**, **13**)/fish	Peptide ^f^	*K. pneumonia* and *A. baumannii* inhibition *in vitro*	1.5–3.1 μM ^+^	Undetermined	TWN	[33]
Antibacterial	adametizine A (**14**)/fungus	Terpenoid ^e^	*S. aureus* inhibition	8 μg/mL ^+^	Undetermined	CHN	[34]
Antibacterial	agelamadins A and B (**15**, **16**)/sponge	Alkaloid ^f^	*M. luteus* and *C. neoformans* inhibition	5–8 μg/mL ^+^	Undetermined	AUS, JPN	[35]
Antibacterial	*Aspergillus* sp. butyrolactone (**17**)/fungus	Terpenoid ^e^	*S. aureus* and *B. cereus* inhibition	1.56 μM ^+^	Undetermined	CHN	[36]
Antibacterial	aszonapyrone A (**18**)/fungus	Terpenoid ^e^	*S. aureus* and *B. subtilis* inhibition	8 μg/mL ^+^	Undetermined	PRT, THAI	[37]
Antibacterial	austalide R (**19**)/fungus	Terpenoid ^e^	Marine bacteria inhibition	0.1 μg/mL ^+^	Undetermined	CHN, DEU, GBR	[38]
Antibacterial	citrifelin B (**20**)/fungus	Polyketide ^d^	*S. aureus* inhibition	4 μg/mL ^+^	Undetermined	CHN	[39]
Antibacterial	desmethylisaridin C1 (**21**)/fungus	Peptide ^f^	*E. coli* inhibition	8 μg/mL	Undetermined	CHN	[40]
Antibacterial	*D. granulosa* diphenyl ethers (**22**, **23**)/sponge	Polyketide ^d^	Gram-positive and negative inhibition	1–16 μg/mL ^+^	Undetermined	USA	[41]
Antibacterial	diaporthalasin (**24**) /fungus	Terpenoid ^e^	MR *S. aureus* inhibition	2 μg/mL ^+^	Undetermined	THAI	[42]
Antibacterial	*D. pulchra* furanones (**25**, **26**)/alga	Alkaloid ^f^	*P. aeruginosa* biofilm inhibition	1.3 μM ^+^	Undetermined	BRA, FRA, USA	[43]
Antibacterial	aureol B (**27**)/ sponge	Terpenoid ^e^	Gram-positive and negative inhibition	1 μg/mL ^+^	Undetermined	S. KOR	[44]
Antibacterial	dysidinoid A (**28**)/sponge	Terpenoid ^e^	MR *S. aureus* inhibition	8 μg/mL **	Undetermined	CHN	[45]
Antibacterial	*Eunicea* sp. compounds (**29**, **30**)/sponge	Terpenoid ^e^	*P. aeruginosa* and *S. aureus* biofilm inhibition	0.5 mg/mL ^+^	Undetermined	COL	[46]
Antibacterial	flavipesin A (**31**)/fungus	Polyketide ^d^	*S. aureus* and *B. subtillis* inhibition	0.25–8 μg/mL ^+^	Undetermined	CHN	[47]
Antibacterial	gageopeptides A–D (**32–35**)/bacterium	Peptide ^f^	*S. aureus* and *B. subtillis* inhibition	0.04–0.08 μM ^+^	Undetermined	BGD, S. KOR	[48]
Antibacterial	gageotetrins A–C (**36–38**)/bacterium	Peptide ^f^	*S. aureus* and *B. subtillis* inhibition	0.02–0.04 μM ^+^	Undetermined	BGD, S. KOR	[49]
Antibacterial	hormaomycin B (**39**)/bacterium	Peptide ^f^	*S. aureus* and *K. rhizophila* inhibition	0.4–7 μM ^+^	Undetermined	S. KOR	[50]
Antibacterial	ieodoglucomide C (**40**)/bacterium	Glycolipid	Gram-positive and negative inhibition	0.01–0.05 μM ^+^	Undetermined	S. KOR	[51]
Antibacterial	isoikarugamycin (**41**)/bacterium	Alkaloid ^f^/ Terpenoid ^e^	MR *S.aureus*	2–4 μg/mL ^+^	Undetermined	ESP	[52]
Antibacterial	keramadine (**42**)/sponge	Alkaloid ^f^	*M. luteus* inhibition	4 μg/mL ^+^	Undetermined	AUS, JPN	[53]
Antibacterial	*Ircinia* sp. secosterols (**43**, **44**)/sponge	Terpenoid ^e^	*M. luteus* and *S. epidermidis* inhibition	3.1, 6.3 μg/mL	Undetermined	S. KOR	[54,55]
Antibacterial	*L. dendyi* terpenoids (**45**, **46**)/sponge	Polyketide ^d^	MR *S. aureus* inhibition	0.05–0.29 μM	Undetermined	USA	[56]
Antibacterial	lindgomycin (**47**)/fungus	Polyketide ^d^	MR *S. aureus* inhibition	5.1 μM	Undetermined	CHN, DEU	[57]
Antibacterial	marformysin D (**48**)/bacterium	Peptide ^f^	*M. luteus* inhibition	0.063 μg/mL ^+^	Undetermined	CHN	[58]
Antibacterial	mollemycin A (**49**)/bacterium	Polyketide ^d^	*S. aureus and S. epidermidis* inhibition	0.05 μM	Undetermined	AUS	[59]
Antibacterial	neolaurene (**50**)/alga	Terpenoid ^e^	*S. typhi* and *S. aureus* inhibition	7.5 μg/mL	Undetermined	MYS	[60]
Antibacterial	penicyclone A (**51**)/fungus	Polyketide ^d^	*S. aureus* inhibition	0.3 μg/mL ^+^	Undetermined	CHN	[61]
Antibacterial	*P. oxalicum* enamide (**52**)/fungus	Polyketide ^d^	*S. aureus* inhibition	2 μg/mL ^+^	Undetermined	CHN	[62]
Antibacterial	puupehenol (**53**)/ sponge	Terpenoid ^e^	*B. cereus* and *S. aureus* inhibition	10 μg/disk ^+^	Undetermined	AUS, USA	[63]
Antibacterial	phyllospongin E (**54**)/sponge	Terpenoid ^e^	*B. cereus* and *S. aureus* inhibition	2.5–3.3 μg/mL ^+^	Undetermined	EGY, GBR	[64]
Antibacterial	sarcotrocheliols (**55**, **56**)/soft coral	Terpenoid ^e^	MR *S. aureus* inhibition	1.5–4.3 μM ^+^	Undetermined	SAU	[65]
Antibacterial	spiromastixone J (**57**)/fungus	Polyketide ^d^	MR *S. aureus* inhibition	2 μM	Undetermined	CHN, DEU	[66]
Antibacterial	stachyin B (**58**)/fungus	Alkaloid ^f^ /Terpenoid ^e^	MR *S. aureus* and *B. subtillis* inhibition	1.4–1.7 μM	Undetermined	CHN, DEU	[67]
Antibacterial	*Streptomyces* sp. glycoside (**59**)/bacterium	Polyketide ^d^	*C. trachomatis* inhibition	4.03 μM	Undetermined	EGY, DEU	[68]
Antibacterial	subergosterones A–C (**60–62**)/gorgonian coral	Terpenoid ^e^	*B. cereus* inhibition	1.6–3.1 μM ^+^	Undetermined	CHN	[69]
Antibacterial	vitroprocine A (**63**)/bacterium	Polyketide ^d^	*A. baumannii* inhibition	8 μg/mL ^+^	Undetermined	TWN, USA	[70]
Antibacterial	xestospongiamide (**64**)/sponge	Polyketide ^d^	Gram-positive and negative inhibition	2.5 μM ^+^	Undetermined	EGY, SAU	[71]
**Antifungal**	bahamaolide A (**65**)/bacterium	Polyketide ^d^	*C. albicans* inhibition	1.5–3.1 μg/mL ^+^	ICL inhibition	S. KOR	[72]
Antifungal	heronamide C (**66**)/bacterium	Polyketal/ alkaloid ^f^	*S. pombe* cell inhibition	5.8 μM ^+^	Alteration of membrane microdomains	JPN	[73]
Antifungal	forazoline A (**67**)/bacterium	Polyketide ^d^	*C. albicans* inhibition	16 μg/mL ^+^	Affected membrane integrity	USA	[74]
Antifungal	aaptamine derivative (**68**)/sponge	Alkaloid ^f^	*T. rubrum* inhibition	4 μg/mL ^+^	Undetermined	CHN	[75]
Antifungal	amphidinin G (**69**)/dinoflagellate	Polyketide ^d^	*T. mentagrophytes* inhibition	8 μg/mL	Undetermined	JPN	[76]
Antifungal	amphidinol 18 (**70**)/dinoflagellate	Polyketide ^d^	*C. albicans* inhibition	9 μg/mL ^+^	Undetermined	ITA	[77]
Antifungal	crambescin homologues (**71–73**)/sponge	Alkaloid ^f^	*C. neoformans* var. *gattii* inhibition	0.85–2.6 μM ^+^	Undetermined	USA	[78]
Antifungal	coustesides C and D (**74**, **75**)/sea cucumber	Terpenoid glycoside ^e^	*C. albicans* inhibition	1 mg/mL ^++^	Undetermined	EGY, S.KOR	[79]
Antifungal	*L. okamurai* laurenes (**76–78**)/alga	Terpenoid ^e^	*C. glabrata* inhibition	2–4 μg/mL **	Undetermined	CHN	[80,81]
Antifungal	mohangamide A (**79**)/bacterium	Peptide ^f^	*C. albicans* isocitrate lyase inhibition	4.4 μM	Undetermined	S. KOR	[82]
Antifungal	pleosporallin E (**80**)/fungus	Polyketide ^d^	*C. albicans* inhibition	7.44 μg/mL ^+^	Undetermined	CHN	[83]
Antifungal	*S. purpurea* lysophospholipid (**81**)/sponge	Phospholipid	*C. glabrata* and *C. neoformans* inhibition	4 μg/mL ^+^	Undetermined	CHN	[84]
Antifungal	taurospongin A (**82**)/sponge	Polyketide ^d^	*C. neoformans* inhibition	1 μg/mL ^+^	Undetermined	AUS, JPN	[85]
Antifungal	variegatuside D (**83**)/sea cucumber	Terpenoid glycoside ^e^	Several *Candida* species inhibition	3.4–13.6 μg/mL ^+^	Undetermined	CHN	[86]
Antifungal	xestospongiamide (**64**)/sponge	Polyketide ^d^	*A. niger* and *C. albicans* inhibition	>5 μM ^+^	Undetermined	EGY, SAU	[71]
**Antimalarial**	*C. hooperi* isonitrile (**84**)/sponge	Terpenoid ^e^	*P. falciparum* D6 and W2 strain inhibition	4.3–4.7 nM	β-hematin inhibition	USA, ZAF	[87]
Antimalarial	actinoramide A (**85**)/bacterium	Peptide ^f^	*P. falciparum* strains inhibition	0.2 μM	Undetermined	CRI, USA	[88]
Antimalarial	diacarperoxide J (**86**)/sponge	Terpenoid ^e^	*P. falciparum* D6 and W2 strain inhibition	1.6–1.8 μM	Undetermined	CHN, USA	[89]
Antimalarial	laevigatol A (**87**)/soft coral	Terpenoid ^e^	*P. falciparum* NF54 strain inhibition	3.0 μM	Undetermined	CHE, DEU, S. KOR, VNM	[90]
Antimalarial	mollemycin A (**49**)/bacterium	Polyketide ^d^	*P. falciparum* 3D7and Dd2 strain inhibition	7–9 nM	Undetermined	AUS	[59]
Antimalarial	mon amphilectines B and C (**88**, **89**)/sponge	Terpenoid ^e^	*P. falciparum* 3D7strain inhibition	44 nM	Undetermined	USA	[91]
Antimalarial	netamine K (**90**)/sponge	Alkaloid ^f^	*P. falciparum* inhibition	2.4 μM	Undetermined	BEL, FRA, ISR	[92]
Antimalarial	*P. ocellata* sesquiterpenes (**91–93**)/nudibranch	Terpenoid ^e^	*P. falciparum* inhibition	0.26–0.3 μM	Undetermined	AUS, ITA	[93]
Antimalarial	*P. simplex* polyketide (**94**)/sponge	Polyketide ^d^	*P. falciparum* D10 and W2 strain inhibition	2.7–4.0 μM	Undetermined	CHN, ITA	[94]
**Antiprotozoal**	plakortide E (**95**)/sponge	Polyketide ^d^	*T. brucei* inhibition	5 μM	Rhodesain inhibition	EGY, DEU	[95]
Antiprotozoal	batzelladine L (**96**)/sponge	Alkaloid ^f^	*T. cruzi* and *L. infantum* inhibition	2 μM	Enhanced ROS generation	BRA, CAN	[96]
Antiprotozoal	actinoporin A (**97**)/ bacterium	Polyketide ^d^	*T. b. brucei* inhibition	15 μM	Undetermined	AUS, DEU, EGY, GBR	[97]
Antiprotozoal	astropectenol A (**98**)/soft coral	Terpenoid ^e^	*T. brucei* inhibition	1.6 μM	Undetermined	DEU, VNM, S. KOR	[98]
Antiprotozoal	*H. simulans* sterol (**99**) sponge	Terpenoid ^e^	*T. b. brucei* inhibition	4.6 μM ^+^	Undetermined	IRL, GBR	[99]
Antiprotozoal	lobosamide A (**100**)/bacterium	Alkaloid ^f^	*T. b. brucei* inhibition	0.8 μM	Undetermined	USA	[100]
Antiprotozoal	lobocrasols A and C (**101**, **102**)/soft corals	Terpenoid ^e^	*L. donovani* inhibition	0.18 μM	Undetermined	CHE, DEU, S. KOR, VNM	[90]
Antiprotozoal	mangromicin A (**103**)/fungus	Polyketide ^d^	*T. b. brucei* inhibition	2.44 µg/mL	Undetermined	JPN	[101]
Antiprotozoal	crassumols D and E (**104**, **105**)/soft corals	Terpenoid ^e^	*T. b. rhodesiense* inhibition	0.61 and 0.72 μM	Undetermined	CHE, DEU, S. KOR, VNM	[90]
Antiprotozoal	sesterstamide (**106**)/sponge	Terpenoid ^e^	*L. donovani* inhibition	32.9 µg/mL	Undetermined	CHN	[102]
Antiprotozoal	shagene A (**107**)/soft coral	Terpenoid ^e^	*L. donovani* inhibition	5 μM	Undetermined	AUS, USA	[103]
**Antituberculosis**	aaptamine analog (**108**)/sponge	Alkaloid ^f^	*M. smegmatis* inhibition	6.25 μg/mL ^+^	Undetermined	JPN	[104]
Antituberculosis	callyaerins A and B (**109**, **110**)/sponge	Peptide ^f^	*M. tuberculosis* inhibition	2, 5 μM **	Undetermined	CHN, DEU, NLD	[105]
Antituberculosis	denigrin C (**111**)/sponge	Alkaloid ^f^	*M. tuberculosis* H_37_Rv inhibition	4 μg/mL ^+^	Undetermined	IND	[106]
Antituberculosis	oxazinin A (**112**)/fungus	Polyketide ^d^	*M. tuberculosis* inhibition	2.9 μM	Undetermined	USA	[107]
**Antiviral**	pateamine A (**113**)/sponge	Mixed Biogenesis	Sindbis virus mRNA translation inhibition	>100 nM	nsP1 or nsP2 viral protein synthesis inhibition	CAN, ESP, NZL	[108]
Antiviral	abyssomicin 2 (**114**)/bacterium	Polyketide ^d^	HIV-1 reactivation	13.9 μM	Increased viral RNA in CD4^+^ T cells	USA	[109]
Antiviral	8,4′’’-dieckol (**115**)/alga	Polyketide ^d^	HIV-1 inhibition	10 μM *	Reverse transcriptase inhibition	S. KOR	[110]
Antiviral	truncateol M (**116**)/fungus	Terpenoid ^e^	H1N1 influenza A virus inhibition	8.8 μM	Virion assembly/release inhibition	CHN, DEU	[111]
Antiviral	neoechinulin B (**117**)/fungus	Alkaloid ^f^	H3N2, H1N1 A influenza virus inhibition	17-22 μM	Hemagglutinin inhibition	CHN, DEU	[112]
Antiviral	thaixylomolin I (**118**)/mangrove	Terpenoid ^e^	H1N1 influenza A virus inhibition	77 μM	Undetermined	CHN, DEU, THAI	[113]
Antiviral	aaptamine derivative (**68**)/sponge	Alkaloid ^f^	HIV-1 inhibition	10 μM *	Undetermined	CHN	[75]
Antiviral	aflaquinolone B derivative (**119**)/fungus	Mixed biogenesis	RSV inhibition	0.042 μM	Undetermined	CHN	[114]
Antiviral	*A. terreus* lactones (**120**, **121**)/fungus	Polyketide ^d^	HSV-1 inhibition	6.34 μg/mL	Undetermined	CHN	[115]
Antiviral	chartarutine B (**122**)/fungus	Alkaloid ^f^/terpenoid ^e^	HIV-1 inhibition	4.9 μM	Undetermined	CHN, DEU	[116]
Antiviral	debromoaplysiatoxin (**123**)/cyanobacterium	Polyketide ^d^	CHIKV inhibition	1.4 μM	Undetermined	NZL, SGP	[117]
Antiviral	dolabelladienol A (**124**)/alga	Terpenoid ^e^	HIV-1 inhibition	2.9 μM	Undetermined	BRA, COL, ESP	[118]
Antiviral	*D. plectens* diterpene (**125**)/alga	Terpenoid ^e^	HIV-1 inhibition	16.1 μM	Undetermined	CHN	[119]
Antiviral	*Dysidea* sp. PBDEs (**22**, **23**)/sponge	Polyketide ^d^	Hepatitis B inhibition	0.23–0.80 μM	Core promoter inhibition	IDN, JPN, NLD	[120]
Antiviral	echrebsteroid C (**126**)/gorgonian	Terpenoid ^e^	RSV inhibition	0.19 μM	Undetermined	CHN	[121]
Antiviral	(+)-pestaloxazine A (**127**)/fungus	Alkaloid ^f^	Enterovirus 71 inhibition	14.2 μM	Undetermined	CHN	[122]
Antiviral	phlorofucofuroeckol-A(**10**)/alga	Polyketide ^d^	MNV inhibition	0.9 μM	Undetermined	S. KOR	[123]
Antiviral	secocrassumol (**128**)/soft coral	Terpenoid ^e^	HCMV inhibition	5 μg/mL	Undetermined	TWN	[124]
Antiviral	sporolide B (**129**)/bacterium	Polyketide ^d^	HIV-reverse transcriptase inhibition	14 μM	Undetermined	IND	[125]
Antiviral	stellettapeptins A and B (**130**, **131**)/sponge	Peptide ^f^	HIV-1 infection inhibition	23–27 nM	Undetermined	USA	[126]
Antiviral	trichobotrysin A (**132**)/fungus	Polyketide ^d^ /Alkaloid ^f^	HSV-1 inhibition	3.08 μM	Undetermined	CHN	[127]
**Anthelmintic**	phorioadenine A (**133**)/sponge	Alkaloid ^f^	*H. contortus* inhibition	31 μg/mL ^+++^	Undetermined	AUS	[128]

**^a^ Organism**: *Kingdom Animalia*: ascidian, flatfish, sea snakes (Phylum Chordata), gorgonian, coral (Phylum Cnidaria), sea cucumber (Phylum Echinodermata), nudibranch (Phylum Mollusca), sponge (Phylum Porifera); *Kingdom Monera*: bacterium (Phylum Cyanobacteria); *Kingdom Fungi*: fungus; *Kingdom Plantae:* alga, mangrove, seagrass; *Kingdom Protista*: dinoflagellates; **^b^ IC_50_**: concentration of a compound required for 50% inhibition in vitro, *: estimated IC_50_, **: MIC_80_, MIC_90_, or IC_90_, ***: in vivo study; ^+^ MIC: minimum inhibitory concentration, ^++^ MID: minimum inhibitory concentration per disk; ^+++^ LC_90_: concentration of a compound required for 90% lethality; **^b^ MMOA**: molecular mechanism of action; **^c^ Country**: AUS: Australia; BEL: Belgium; BGD: Bangladesh; BRA: Brazil; CAN: Canada; CHE: Switzerland; CHN: China; COL: Colombia; CRI: Costa Rica; DEU: Germany; EGY: Egypt; ESP: Spain; FRA: France; GBR: United Kingdom; IDN: Indonesia; IND: India; IRL: Ireland; ISR: Israel; ITA: Italy; JPN: Japan; MYS: Malaysia; NLD: The Netherlands; NZL: New Zealand; PRT: Portugal; SAU: Saudi Arabia; SGP: Singapore; S. KOR: South Korea; THAI: Thailand; TWN: Taiwan; VNM: Vietnam; ZAF: S. Africa; **Chemistry: ^d^** polyketide; **^e^** terpene; **^f^** nitrogen-containing compound; **^g^** polysaccharide; **^h^** shikimate; **Abbreviations:** CHIKV: chikungunya virus; HCMV: human cytomegalovirus; MNV: murine norovirus; HSV: herpes simplex virus; ICL: isocitrate lyase; MR: methicillin-resistant; PBP2a: penicillin-binding protein 2a; RNAP: RNA-polymerase; RSV: respiratory syncytial virus; TNF-α: tumor necrosis factor α.

**Table 2 marinedrugs-18-00005-t002:** Marine pharmacology in 2014–2015: marine compounds with antidiabetic and anti-inflammatory activity, and affecting the immune and nervous system.

Drug Class	Compound/Organism ^a+^	Chemistry	Pharmacological Activity	IC_50_ ^b^	MMOA ^c^	Country ^d^	References
Antidiabetic	fucoxanthin and fucoxanthinol (**134**, **135**)/alga	Terpenoid ^f^	Improved glucose tolerance in vitro and in vivo	50 µM *	Cytokine inhibition	JPN, S. KOR	[154,155]
Antidiabetic	octaphlorethol A (**136**)/alga	Shikimate ^h^	α-glucosidase inhibition	110 µM	Molecular docking on active site	CAN, S. KOR	[156]
Antidiabetic	phlorofucofuroeckol-A (**10**)/alga	Polyketide ^d^	Decreased glucose levels in vivo	10 mg/kg **	α-glucosidase inhibition	S. KOR	[157]
Antidiabetic	Con-Ins G1 (**137**)/cone snail	Peptide ^g^	Hypoglycemia induction	65 ng/g *	Undetermined	AUS, DNK, USA	[158]
Antidiabetic	dehydroeuryspongin A (**138**)/sponge	Terpenoid ^f^	PTP1B inhibition	3.58 μM	Undetermined	IDN, JPN	[159]
Antidiabetic	*Epicoccum* sp. diterpene (**139**)/fungus	Terpenoid ^f^	α-glucosidase inhibition	4.6 μM	Undetermined	CHN	[160]
Antidiabetic	suncheonoside A (**140**)/bacterium	Terpenoid ^f^	Adiponectin production	10 μM *	Undetermined	S. KOR	[161]
Antidiabetic	terrelumamide A (**141**)/fungus	Peptide ^g^	Adiponectin production	37 μM *	Undetermined	S. KOR	[162]
Antidiabetic	*X. testudinaria* lipid (**142**)/sponge	Polyketide ^d^	PTP1B inhibition	5.3 μM	Undetermined	CHN	[163]
Anti-inflammatory	alcyonolide congeners (**143**, **144**)/soft coral	Terpenoid ^f^	Macrophage NO inhibition	2 μM *	iNOS expression inhibition	JPN	[164]
Anti-inflammatory	astaxanthin (**145**)/alga	Terpenoid ^f^	Oxidative stress inhibition in vivo	10 mg/kg **	CAT and SOD enhancement	CHN	[165]
Anti-inflammatory	8,8′-bieckol (**146**)/alga	Polyketide ^e^	Macrophage NO and PGE_2_ release inhibition	50 μM *	Inhibition of NFκB	S. KOR	[166]
Anti-inflammatory	convolutamydine A (**147**)/ bryozoa	Alkaloid ^g^	Formalin-induced licking behavior inhibition	0.01 mg/kg *	TNF-α, IL-6 release inhibition	BRA	[167]
Anti-inflammatory	capgermacrene A (**148**)/ soft coral	Terpenoid ^f^	Macrophage NO and IL-1β inhibition	<10 μg/mL *	iNOS expression inhibition	MYS, S. KOR	[168]
Anti-inflammatory	cathelicidin (**4**)/sea snake	Peptide ^g^	Binding of LPS to TLR4 inhibition	4 μg/mL *	Inflammatory cytokine inhibition	CHN	[26]
Anti-inflammatory	dactyloditerpenol acetate (**149**)/ sea hare	Terpenoid ^f^	LPS- activated microglia in vitro inhibition	0.4–1 μM	O_2_^-^ and TXB_2_ inhibition	USA	[169]
Anti-inflammatory	dieckol (**150**)/alga	Shikimate ^h^	Macrophage iNOS transcription inhibition	30 μM *	Inhibition of NFκB and p38MAPK	S. KOR	[170]
Anti-inflammatory	dieckol (**150**)/alga	Shikimate ^h^	Human keratinocyte MDC/CCL22 inhibition	12.5 μM *	STAT1 phosphorylation inhibition	S. KOR	[171]
Anti-inflammatory	excavatolide B (**151**)/gorgonian	Terpenoid ^f^	Macrophage iNOS and COX-2 transcription inhibition	25 μM *	In vivo iNOS protein expression reduction	TWN	[172]
Anti-inflammatory	flexibilide (**152**)/soft coral	Terpenoid ^f^	Neuropathic pain inhibition	10 µg *	Upregulation of TGF-β1	TWN	[173]
Anti-inflammatory	fucoxanthinol (**135**)/alga	Terpenoid ^f^	Macrophage TNF-α and MCP-1 release inhibition	10 μM	COX-2 expression inhibition	JPN	[155]
Anti-inflammatory	*H. fusiforme* flavone (**153**)/ alga	Shikimate ^h^/Polyketide ^d^	Macrophage NO and PGE_2_ release inhibition	10 μg/mL *	iNOS, COX-2 expression inhibition	S. KOR	[174]
Anti-inflammatory	5β-hydroxypalisadin B (**154**)/alga	Terpenoid ^f^	Macrophage NO release inhibition	17 μM	Partial iNOS expression inhibition	LKA, MYS, S. KOR	[175]
Anti-inflammatory	glaucumolides A and B (**155**, **156**)/soft coral	Terpenoid ^f^	Neutrophil SOX and elastase inhibition	2.8–4 µM *	iNOS, COX-2 inhibition	TWN	[176]
Anti-inflammatory	phlorofucofuroeckol-B (**157**)/alga	Polyketide ^d^	Microglia activation inhibition	0.1 μg/mL *	iNOS, COX-2 inhibition	S. KOR	[177]
Anti-inflammatory	*P. palmata* lipid (**158**)/alga	Polyketide ^d^	Macrophage NO release inhibition	16.7 μM	iNOS expression inhibition	CAN	[178]
Anti-inflammatory	reduced scytonemin (**159**)/alga	Alkaloid ^g^	Macrophage NO release inhibition	1 µM *	HO-1 expression induction	JPN	[179]
Anti-inflammatory	sinuleptolide (**160**)/soft coral	Terpenoid ^f^	LPS-activated rat microglia in vitro inhibition	0.5–2.9 μM	Cytokine release inhibition	ESP, FIN, IND, ITA,	[180]
Anti-inflammatory	sarcopanol A (**161**)/soft coral	Terpenoid ^f^	iNOS, COX-2, and ICAM-1 transcription inhibition	8.3 µM	NFκB inhibition	S. KOR, VNM	[181]
Anti-inflammatory	sinumaximol H (**162**)/sponge	Terpenoid ^f^	iNOS and ICAM-1 transcription inhibition	1 µM *	NFκB inhibition	S. KOR, VNM	[182]
Anti-inflammatory	tanzawaic acid A (**163**)/fungus	Polyketide ^d^	NO inhibition	7.1 µM	iNOS and PTP1B inhibition	VNM, S. KOR	[183]
Anti-inflammatory	aspertetranone D (**164**)/fungus	Terpenoid ^f^	IL-6 inhibition	40 µM *	Undetermined	CHN, USA	[184]
Anti-inflammatory	briarenolide J (**165**)/soft coral	Terpenoid ^f^	Neutrophil SOX and elastase inhibition	10–15 µM	Undetermined	TWN	[185]
Anti-inflammatory	briarenolides K and L (**166**, **167**)/soft coral	Terpenoid ^f^	Macrophage iNOS inhibition	>10 μg/mL *	Undetermined	TWN	[186]
Anti-inflammatory	briarenolides U, V, W (**168**, **169**)/soft coral	Terpenoid ^f^	Macrophage COX-2 and iNOS expression inhibition	>10 μg/mL *	Undetermined	TWN	[187]
Anti-inflammatory	briaviolides E and I (**170**, **171**)/soft coral	Terpenoid ^f^	Neutrophil SOX and elastase inhibition	>10 μg/mL *	Undetermined	TWN	[188]
Anti-inflammatory	dermacozine H (**172**)/bacterium	Alkaloid ^g^	Radical scavenging activity	18.8 µM	Undetermined	DEU, EGY, UK	[189]
Anti-inflammatory	dysifragilone A (**173**)/sponge	Terpenoid ^f^	Macrophage NO release inhibition	6.6 µM	Undetermined	CHN	[190]
Anti-inflammatory	*D. plectens* xenicane (**174**)/alga	Terpenoid ^f^	Macrophage NO release inhibition	10 µM	Undetermined	CHN	[191]
Anti-inflammatory	comaparvin (**175**)/crinoid	Polyketide ^d^	Carrageenan-induced hyperalgesia inhibition	30 mg/kg *	iNOS expression inhibition	TWN	[192]
Anti-inflammatory	hirsutalins N and S (**176**, **177**)/ soft coral	Terpenoid ^f^	Neutrophil elastase inhibition	10 μM *	Undetermined	TWN	[193,194]
Anti-inflammatory	hirsutocospiro A (**178**)/soft coral	Terpenoid ^f^	Neutrophil SOX and elastase inhibition	3.7–4.1 μM	Undetermined	TWN	[195]
Anti-inflammatory	isosinulaflexiolide K (**179**)/soft coral	Terpenoid ^f^	Macrophage COX-2 and iNOS expression inhibition	>10 μM *	Undetermined	TWN	[196]
Anti-inflammatory	klyflaccisteroid F (**180**)/soft coral	Terpenoid ^f^	Neutrophil SOX and elastase inhibition	0.34 μM	Undetermined	TWN	[197]
Anti-inflammatory	krempfielin N (**181**)/soft coral	Terpenoid ^f^	Neutrophil SOX and elastase inhibition	>10 μM *	Undetermined	TWN	[198]
Anti-inflammatory	krempfielins Q and R (**182**, **183**)/soft coral	Terpenoid ^f^	Neutrophil SOX and elastase inhibition	>10 μM *	Undetermined	TWN	[199]
Anti-inflammatory	methylfarnesylquinone (**184**)/alga	Shikimate ^h^/Terpenoid ^f^	Neutrophil SOX and elastase inhibition	0.2–0.48 µg/mL	Undetermined	TWN	[200]
Anti-inflammatory	monanchosterol B (**185**)/sponge	Terpenoid ^f^	Macrophage IL-6 expression inhibition	5 µM	Undetermined	S. KOR	[201]
Anti-inflammatory	*P. nodosus* sterol (**186**)/starfish	Terpenoid ^f^	IL-12 and IL-6 inhibition	1.3–3.1 µM	Undetermined	VNM, S. KOR	[202]
Anti-inflammatory	rhytidenone C (**187**)/fungus	Polyketide ^d^	Macrophage NO inhibition	0.31 µM	Undetermined	THA	[203]
Anti-inflammatory	sarcocrassocolide E (**188**)/soft coral	Terpenoid ^f^	Macrophage COX-2 and iNOS expression inhibition	<10 μM *	Undetermined	TWN	[204]
Anti-inflammatory	sinulacembranolide A (**189**)/octocoral	Terpenoid ^f^	Macrophage iNOS expression inhibition	<10 μM	Undetermined	TWN	[205]
Anti-inflammatory	thomimarine B (**190**)/fungus	Terpenoid ^f^	Macrophage NO inhibition	>10 μM	Undetermined	RUS, VNM	[206]
Anti-inflammatory	tortuosene A (**191**)/soft coral	Terpenoid ^f^	Neutrophil SOX inhibition	7.3 μM	Undetermined	TWN	[207]
Immune system	grassypeptolide A (**192**)/cyanobacterium	Peptide ^g^	IL-2 and T-cell proliferation inhibition	1 μM *	Dipeptidyl peptidase 8 inhibition	CHN, JPN, USA	[208]
Immune system	*F. reticulata* alkaloids (**193**)/sponge	Alkaloid ^g^	IL-2 inhibition	5–50 µM *	Undetermined	CHN, NLD	[209]
Immune system	luzonicoside A (**194**)/starfish	Terpenoid ^f^	Macrophage NO and ROS stimulation	0.01–0.1 µM *	Undetermined	RUS, VNM	[210]
Immune system	typicoside C1(**195**)/sea cucumber	Terpenoid ^f^	Macrophage ROS stimulation	<1 ng/mL *	Undetermined	IND, RUS	[211]
**Nervous system**	aurone glycoside (**196**)/fungus	Shikimate ^h^/Polyketide ^d^	Oxidative stress neuroprotection	1 µM *	Apoptosis inhibition	CHN	[212]
Nervous system	azaspiracid-1 (**197**)/alga	Polyketide ^d^/ Alkaloid ^g^	Peripherin-labelled neurite process	15 nM *	Peripherin isoform downregulation	NOR	[213]
Nervous system	caulerpine (**198**)/alga	Alkaloid ^g^	Antinociceptive activity	40 mg/kg *	Involves α2 and 5-HT_3_ receptors	BRA	[214]
Nervous systema	6-bromohypaphorine (**199**)/sea slug	Alkaloid ^g^	Human α7 nAChR agonist	23 µM	Rise [Ca^2+^ ]i	RUS	[215]
Nervous system	piscidin (**200**)/fish	Peptide ^g^	Antinociceptive activity	20 µg/rat *	Phosphor-mTOR inhibition	TWN	[216]
Nervous system	*C. marmoreus* conotoxin Mr1.7 (**201**)/cone snail	Peptide ^g^	Ach-evoked membrane current inhibition	53.1 nM	A3β2 nAChR inhibition	CHN	[217]
Nervous system	*C. litteratus* conotoxin lt16a (**202**)/cone snail	Peptide ^g^	Neuronal Na^+^ current inhibition	1 µM *	Undetermined	CHN	[218]
Nervous system	*C. vitulinus* peptide (**203**)/ cone snail	Peptide ^g^	Neuronal BK channel inhibition	8.5 µM	Electrostatic interaction with β4 subunits	CHN, USA	[219]
Nervous system	echinochrome A (**204**)/sea urchin	Polyketide ^d^	Acetylcholinesterase inhibition and NO scavenging	16.4 µM	Irreversible and uncompetitive inhibition	S. KOR, RUS	[220]
Nervous system	ganglioside LLG-3 (**205**)/starfish	Glycolipid	Neuritogenesis stimulation in vitro	1 nM *	MAPK signaling stimulation	JPN	[221]
Nervous system	heteronemin (**206**)/sponge	Terpenoid ^f^	TDP-43 binding to DNA inhibition	10.1 nM	Promoted aggregation of insoluble TDP-43	ITA	[222]
Nervous system	pinnatoxin A (**207**)/mollusc	Polyketide ^d^/ Alkaloid ^g^	Muscle and neuronal nAChRs receptor inhibition	0.086–47.5 nM	EF-ketal ring confers nAChR subtype specificity	FRA, USA	[223]
Nervous tissue	PhcrTx1 (**208**)/sea anemone	Peptide ^g^	ASIC inhibition	100 nM	Lower potency on Kv channels	BEL, BRA, CUB, DEU, ESP, MEX	[224]
Nervous tissue	phlorofucofuroeckol-A (**10**)/alga	Polyketide ^d^	Butyrylcholinesterase inhibition	0.95 µM	β-secretase inhibition	S. KOR	[225]
Nervous tissue	*S. auritum* ceramide (**209**)/soft coral	Polyketide ^d^	Anxiolytic and CNS depressing activity in vivo	1 mg/kg **	GABA(A) receptor modulation	EGY, USA	[226]
Nervous system	spirolide C (**210**)/dinoflagellate	Polyketide ^d^/ Alkaloid ^g^	nAChR inhibition	1.5–3 nM *	Muscle and neuronal -type nAChR inhibition	FRA	[227]
Nervous system	zonarol (**211**)/alga	Meroterpenoid ^f^	Glutamate toxicity inhibition in vitro	0.22 µM	Nrf2/ARE pathway activation	JPN, USA	[228]
Nervous system	aplysinellamide-1 (**212**)/sponge	Alkaloid ^g^	ApoE secretion modulation	30 µM *	Undetermined	AUS, CAN	[229]
Nervous system	*A. terreus* lactones (**120**, **121**)/fungus	Polyketide ^d^	acetylcholinesterase inhibition	4.2 µM	Undetermined	CHN	[115]
Nervous system	*C. araneosus* ar3j conotoxin (**213**)/cone snail	Peptide ^g^	Sleep induction	2 nM *	Undetermined	IND	[230]
Nervous system	*D. cejpii* steroid (**214**)/fungus	Terpenoid ^f^	Amyloid β-42 production inhibition	10 µM *	Undetermined	DEU, FRA	[231]
Nervous system	genuanine (**215**)/cone snail	Alkaloid ^g^	Paralysis in vivo	40 nM *	Undetermined	PRT, USA	[232]
Nervous system	homoaerothionin (**216**)/sponge	Alkaloid ^g^	acetylcholinesterase inhibition	2.9–6.2 µM	Undetermined	THA	[233]
Nervous system	mooreamide A (**217**)/bacterium	Polyketide ^d^	CB_1_ binding	0.47 µM **	Undetermined	ITA, PNG, USA	[234]
Nervous system	*S. spinosulus* hydroquinone (**218**)/sponge	Shikimate ^h^/Polyketide ^d^	Enhance glutamate and ACh release	10 µM *	Undetermined	ITA	[235]

^a^**Organism:***Kingdom Animalia*: fish (Phylum Chordata); bryozoan; coral and sea anemone (Phylum Cnidaria); crinoid, sea urchin, starfish (Phylum Echinodermata); cone snail, sea slug (Phylum Mollusca); sponge (Phylum Porifera); *Kingdom Fungi*: fungus; *Kingdom Plantae:* alga; *Kingdom Monera*: bacterium; *Kingdom Protozoa*: dinoflagellates; ^b^
**IC_50_**: concentration of a compound required for 50% inhibition, *: apparent IC_50_, **: in vivo study; **: *K*_i_: concentration needed to reduce the activity of an enzyme by half; ^c^
**MMOA:** molecular mechanism of action; ^d^
**Country:** AUS: Australia; BEL: Belgium; BRA: Brazil; CAN: Canada; CHN: China; CUB: Cuba; DEU: Germany; DNK: Denmark; EGY: Egypt; ESP: Spain; FIN, Finland; FRA: France; IDN: Indonesia; IND, India; ITA: Italy; JPN: Japan; LKA: Sri Lanka; MEX: Mexico; MYS: Malaysia; NLD: Netherlands; NOR: Norway; PNG: Papua New Guinea; PRT: Portugal; RUS: Russian Federation; S. KOR: South Korea; THA: Thailand; TWN: Taiwan; VNM: Vietnam; **Chemistry:**
^e^ Polyketide; ^f^ Terpene; ^g^ Nitrogen-containing compound; ^h^ Shikimate. **Abbreviations:** Ach: acetylcholine; ApoE: apolipoprotein E; ASIC: acid-sensing sodium ion channel; CAT: catalase; CB_1_: cannabinoid receptor 1; CNS: central nervous system; COX: cyclooxygenase; HO-1: heme oxygenase-1; ICAM: intercellular adhesion molecule-1; IL: interleukin; iNOS: inducible nitric oxide synthase; Kv current: voltage-gated K+ current; MAPK: mitogen-activated protein kinase pathway; MDC/CCL22: macrophage-derived chemokine, C–C motif chemokine 22; nAChR: nicotinic acetylcholine receptor; NA: not available; NF-κB: nuclear factor kappa-light-chain-enhancer of activated B cells; NO: nitric oxide; nAChR: nicotinic acetylcholine receptor; Nrf2-ARE: nuclear transcription factor E2-related factor antioxidant response element; PTP1B: tyrosine protein; phosphatase 1B; ROS: reactive oxygen species; SOD: superoxide dismutase; SOX: superoxide; STAT1: signal transducer and activator of transcription1; TDP-43: trans-activation response DNA-binding protein of 43 kDa.

**Table 3 marinedrugs-18-00005-t003:** Marine pharmacology in 2014–2015: marine compounds with miscellaneous mechanisms of action.

Compound/Organism ^a^	Chemistry	Pharmacological Activity ^i^	IC_50_ ^b^	MMOA ^c^	Country ^d^	References
AdE-1 (**219**)/sea anemone	Peptide ^g^	Cardiomyocyte action potential modulation	2 nM *	Na^+^ and K^+^ current increase	ISR	[236]
aaptamine (**220**)/sponge	Alkaloid ^g^	ROS inhibition	10 μM *	Cytokine inhibition	S. KOR	[237]
amphirionin-4 (**221**)/dinoflagellate	Polyketide ^e^	Bone marrow stromal cells proliferation stimulation	<0.1 ng/mL	Cytoskeleton protein synthesis	JPN	[238]
astaxanthin (**145**)/alga	Terpenoid ^f^	Leydig cell steroidogenesis protection	10 µg/mL *	ROS scavenging	TWN	[239]
bastadins 6 and 16 (**222**, **223**)/sponge	Alkaloid ^g^	Foam cell formation inhibition	5 μM *	ACAT inhibition	JPN	[240]
6,6-bieckol (**224**)/alga	Polyketide ^e^	Adipocyte differentiation inhibition	50 µg/mL *	Adipogenesis inhibition	S. KOR	[241]
(-)-chlorizidine A (**225**)/bacterium	Alkaloid ^g^	Increase G_1_ cell cycle phase	2 μM *	GAPDH and hENO1 binding	BRA, USA	[242]
dihydroaustrasulfone alcohol (**226**)/soft coral	Polyketide ^e^	PDGF-induced HASMC proliferation and angiogenesis inhibition	10 μM *	DNA synthesis and VEGF signaling inhibition	TWN	[243,244]
DPHC (**227**)/alga	Terpenoid ^f^	UVB radiation-induced DNA damage protection	20 μM *	Nucleotide excision repair system induction	S. KOR	[245]
echinochrome A (**204**)/sea urchin	Alkaloid ^g^	Cardiac contractility inhibition	3 μM *	SERCA2A inhibition	BEL, S. KOR, RUS	[246]
echinochrome A (**204**)/sea urchin	Alkaloid ^g^	Increased mitochondria biogenesis and function	5 μM *	Mitochondrial biogenesis genes upregulation	S. KOR, RUS	[247]
eckol (**228**)/alga	Polyketide ^e^	ROS suppression in cells	10 μM *	Increased HO-1 expression	S. KOR	[248]
emindole SB (**229**)/fungus	Terpenoid ^f^	Nonselective CB_1_/CB_2_ antagonist	2.2–7.0 μM **	Undetermined	CHE, DEU	[249]
farnesylquinone (**230**)/bacterium	Polyketide ^e^	Decreased lipid accumulation	1 μM *	Increased PPARα activity	CHN, DEU	[250]
FGFC1 (**231**)/fungus	Alkaloid ^g^	Thrombolysis induction in vivo	5 mg/kg *	Fibrin hydrolysis induction in vitro	CHN	[251]
formosusin A (**232**)/fungus	Alkaloid ^g^	Mammalian DNA polymerase β inhibition	35.6 μM	Competitive and non-competitive inhibition	JPN	[252]
fucodiphlorethol (**233**)/alga	Polyketide ^e^	ROS inhibition	10 μM *	Decreased mitochondrial loss, and caspase-9 expression	S. KOR	[253]
gallic acid (**234**)/alga	Shikimate ^h^	NO-dependent vasorelaxant effect	12.5 µg/mL	Phospho-eNOS increase	S. KOR	[254]
gallinamide A (**235**)/bacterium	Peptide ^f^	Human cathepsin L inhibition	5 nM	Covalent inhibition	USA	[255]
girolline (**236**)/sponge	Alkaloid ^g^	TLR 5 inhibition	2 µg/mL	IL-8 and IL-6 inhibition	CAN, NLD, USA	[256]
gracilins A, H, L (**237–239**)/sponge	Terpenoid ^f^	mPTP opening inhibition	1 μM *	Binding to CypD	EGY, ESP, GBR	[257]
*H. crispa* polypeptides (**240**, **241**)/sea anemone	Peptide ^f^	Macrophage TNF-α, IL-6, and proIL-1β inhibition		Trypsin and α-chemotrypsin inhibition	RUS, TWN	[258]
*Hyrtios* sp. sesterterpene (**242**)/sponge	Terpenoid ^f^	TDP-43 inhibition	0.4 nM	TDP-43 to DNA binding inhibition	ITA, PYF	[259]
irciniastatin A (**243**)/sponge	Polyketide ^e^	TNF-α receptor 1 ectodomain shedding	10 nM *	ERK activation induced	JPN	[260]
iejimalide C (**244**)/ascidian	Polyketide ^e^	V-ATPase inhibitor	0.12 μM	Bafilomycin site binding	JPN	[261]
*Laurencia* sp. indole (**245**)/alga	Alkaloid ^g^	Aryl hydrocarbon receptor agonist	10 μM *	DNA binding stimulation and CYP1A1 induction	JPN, USA	[262]
neopetroside A (**246**)/sponge	Alkaloid ^g^	Cardiomyocyte mitochondrial upregulation	10 µM *	Increased ATP levels and O_2_ consumption	RUS, S. KOR	[263]
Phlorofucofuroeckol-A (**10**)/alga	Polyketide ^e^	Lipid accumulation inhibition	18 µM	Decreased PPARγ expression	S. KOR	[264]
polytheonamide B (**247**)/sponge	Peptide ^g^	One-ion pore channel permeation determined	NA	Two ion binding sites defined, but second ion excluded	JPN	[265]
siphonaxanthin (**248**)/alga	Terpenoid ^f^	Adipogenesis inhibition	5 µM	Transcription factor inhibition	JPN	[266]
spiromastixones J and L (**57**, **249**)/fungus	Polyketide ^e^	Cholesterol uptake inhibition	10 μM *	PPARγ upregulation	CHN	[267]
thalassospiramide C (**250**)/bacterium	Peptide ^g^	HCAN1 inhibition	3.4 nM	Binding to Cys115 residue	CHN, USA	[268]
xyloketal B (**251**)/fungus	Polyketide ^e^	Atherosclerotic plaque attenuation	14 mg/kg ***	Increased eNOS activity	CAN, CHN	[269]
xyloketal B (**251**)/fungus	Polyketide ^e^	P450 3a activity and expression regulation	14 mg/kg ***	Active site binding determined by docking studies	CHN, USA	[270]
*X. testudinaria* lipid (**252**)/sponge	Polyketide ^e^	Pancreatic lipase inhibition	3.11 μM	Triglyceride level decrease in vivo	CHN, ITA	[271]
acuminolide A (**253**)/dinoflagellate	Polyketide ^e^	Stimulation of actomyosin ATPase	1 μM *	Undetermined	S. KOR	[272]
ahpatinin Ac (**254**)/bacterium	Peptide ^g^	Pepsin inhibition	11 nM	Undetermined	JPN	[273]
alternariol derivatives (**255–257**)/sponge	Shikimate ^h^	HCV protease inhibition	12–52 µg/mL	Undetermined	EGY, SAU	[136]
apliamide D (**258**)/ascidian	Peptide ^g^	Na^+^/K^+^-ATPase inhibition	3.2 μM	Undetermined	S. KOR	[274]
austalides 4 and 9 (**259**, **260**)/fungus	Terpenoid ^f^	*Endo*-1,3-*β*-D-glucanase inhibition	0.01 μM	Undetermined	RUS	[275]
axinelline A (**261**)/bacterium	Alkaloid ^g^	COX-2 inhibition	2.8 μM	Undetermined	CHN	[276]
brevisulcatic acid-4 (**262**)/dinoflagellate	Polyketide ^e^	Activation of sodium channels	20 ng/mL	Undetermined	JPN, NZL	[277]
4′,5′-dehydrodiodictyonema (**263**)/alga	Terpenoid ^f^	PTP1B inhibition	2.3 μM	Undetermined	CHN	[278]
*D. metachromia* sesquiterpenes (**264–266**)/sponge	Terpenoid ^f^	Multiple kinases inhibition	0.97–4.8 μM	Undetermined	CHN, NLD, DEU	[279]
didemnaketals D and E (**267**, **268**)/ascidian	Terpenoid ^f^	Multiple kinases inhibition	10 μg/mL *	Undetermined	EGY	[280]
dysiquinol D (**269**)/sponge	Terpenoid ^f^	NF-κB inhibition	0.81 μM	Undetermined	AUS, CHN	[281]
fucoxanthin (**134**)/alga	Terpenoid ^f^	Hydroxyl radical-scavenging	10 μg/mL *	Undetermined	CHN	[282]
halenaquinol sulfate (**270**)/sponge	Polyketide ^e^	CDK9 and DYRK1A inhibition	0.5–0.61 μM	Undetermined	FRA, NZL	[283]
1-hydroxyethylhalenaquinone (**271**)/sponge	Polyketide ^e^	Proteasome-chymotrypsin-like activity inhibition	0.19 μM	Undetermined	IDN, JPN, NLD	[284]
hymenialdisine derivatives (**272–274**)/sponge	Alkaloid ^g^	P*f*GSK-3 inhibition	0.07–0.2 μM	Undetermined	DEU, EGY, FRA, NLD	[285]
hymenialdisine (**275**)/sponge	Alkaloid ^g^	CK1, CDK5, GSK3β inhibition	0.03–0.16 μM	Undetermined	AUS	[286]
hyattellactone A (**276**)/sponge	Terpenoid ^f^	PTP1B inhibition	7.45 μM	Undetermined	IDN, JPN	[287]
*H. lachne* sesterterpenoid (**277**)/sponge	Terpenoid ^f^	PTP1B inhibition	5.2 μM	Undetermined	CHN	[288]
ianthelliformisamines A–C (**278**, **7**, **8**)/ sponge	Alkaloid ^g^	Carbonic anhydrase inhibition	0.2–0.85 μM **	Undetermined	AUS, ITA	[289]
incisterols A5 and A6 (**279**, **280**)/sponge	Terpenoid ^f^	PXR agonists	10 μM *	Undetermined	ITA	[290]
*L. crassum* cembranoid (**281**)/soft coral	Terpenoid ^f^	PPAR transcription activation	2.07 μM	Undetermined	VNM	[291]
*L. okamurai* terpenoid (**282**)/alga	Terpenoid ^f^	PTP1B inhibition	4.9 μg/mL	Undetermined	CHN	[292]
malonganenone L (**283**)/sea whip	Alkaloid ^g^	PDE4D inhibition	8.5 μM	Undetermined	CHN	[293]
punctaporonin K (**284**)/fungus	Terpenoid ^f^	Lipid-lowering effect	10 μM *	Undetermined	CHN, DEU	[294]
psychrophilin G (**285**)/fungus	Peptide ^g^	Lipid-lowering effect	10 μM *	Undetermined	CHN	[295]
racemosin (**286**)/alga	Alkaloid ^g^	PTP1B inhibition	5.9 µM	Undetermined	CHN	[296]
sarcoehrendin B (**287**)/soft coral	Polyketide ^e^	PDE4 inhibition	3.7 µM	Undetermined	CHN	[297]
sarsolilide A (**288**)/soft coral	Terpenoid ^f^	PTP1B inhibition	6.8 µM	Undetermined	CHN, HUN	[298]
*Stachybotry* sp. xanthone (**289**)/fungus	Polyketide ^e^	COX-2 inhibition	8.9 µM	Undetermined	CHN	[299]
*S. thunbergii* alkapolyene (**290**)/alga	Polyketide ^e^	Soybean LOX inhibition	5 µM	Undetermined	JPN, S. KOR	[300]
shinorine (**291**)/alga	Alkaloid ^g^	*C. histolyticum* collagenase inhibition	104 µM	Undetermined	AUT	[301]
sinularone D (**292**)/soft coral	Polyketide ^e^	NF-κB inhibition	10 μg/mL *	Undetermined	CHN	[302]
*N*-(2-benzenepropanoic acid) stachybotrylactam (**293**)/fungus	Alkaloid ^g^	Triglyceride and cholesterol inhibition	10 μM *	Undetermined	CHN, DEU	[303]
strongylophorine-13/-14 (**294**)/sponge	Terpenoid ^f^	Hu proteasome 20S inhibition	2.1 μM	Undetermined	JPN	[304]
swinhoeisterol A (**295**)/sponge	Terpenoid ^f^	(h)P300 acetyltransferase inhibition	2.7 μM	Undetermined	ITA, CHN, USA	[305]
thunberol (**296**)/alga	Terpenoid ^f^	PTP1B inhibition	2.24 μg/mL	Undetermined	CHN	[306]
xestosaprol O (**297**)/sponge	Terpenoid ^f^	IDO1 inhibition	4 μM	Undetermined	CAN, NLD	[307]
urupocidin A (**298**)/sponge	Alkaloid ^g^	iNOS expression induction	10 μM *	Undetermined	RUS, TWN	[308]
variabine B (**299**)/sponge	Alkaloid ^g^	Proteasome-chymotrypsin-like activity inhibition	4 µg/mL	Undetermined	IDN, JPN, NLD	[309]
yoshinone A (**300**)/cyanobacterium	Polyketide ^e^	Triglyceride inhibition	0.4 μM	Undetermined	JPN	[310]

^a^**Organism**, *Kingdom Animalia*: ascidian (Phylum Chordata), soft corals, sea whips, and sea anemone (Phylum Cnidaria), dinoflagellates (Phylum Dinoflagellata), sea urchin (Phylum Echinodermata), sponge (Phylum Porifera); *Kingdom Fungi*: fungus; *Kingdom Plantae:* alga; *Kingdom Monera*: bacterium; ^b^
**IC_50_**: concentration of a compound required for 50% inhibition in vitro; *: estimated IC_50_; **: Ki; *** in vivo study; ^c^
**MMOA**: molecular mechanism of action; ^d^
**Country:** AUS: Australia; AUT: Austria; BEL: Belgium; BRA: Brazil; CAN: Canada; CHE: Switzerland; CHN: China; DEU: Germany; EGY: Egypt; FRA: France; ESP: Spain; GBR: United Kingdom; HUN: Hungary; IDN: Indonesia; ISR: Israel; ITA: Italy; JPN: Japan; NLD: The Netherlands; NZL: New Zealand; PYF: French Polynesia; RUS: Russian Federation; SAU: Saudi Arabia; S. KOR: South Korea; TWN: Taiwan; VNM: Vietnam; **Chemistry:**
^e^ Polyketide; ^f^ Terpene; ^g^ Nitrogen-containing compound; ^h^ shikimate; **Abbreviations:** ACAT: acyl-CoA:cholesterol acyl-transferase; CB: cannabinoid; CDK: cyclin-dependent kinase; COX-2: cyclooxygenase 2; CK1: casein kinase 1; CypD: cyclophilin D; DDYRK: dual-specificity, tyrosine phosphorylation regulated kinase; DPHC: diphlorethohydroxycarmalol; eNOS: endothelial nitric oxide synthase; ERK: extracellular signal-regulated kinase; GAPDH: D-glyceraldehyde-3-phosphate dehydrogenase; GSK3β: glycogen synthase kinase 3; HASMC: human aortic smooth muscle cells; HCAN1: human calpain 1 protease; HCV: hepatitis C virus; hENO1: human alpha-enolase; hu: human; HO-1: hemeoxygenase-1; IDO1: indoleamine 2, 3 dioxygenase; IL: interleukin; iNOS: inducible nitric oxide synthase; LOX: lipoxygenase; NF-κB: nuclear factor kappa-light-chain-enhancer of activated B cells; NO: nitric oxide; mPTP: mitochondrial permeability transition pore; PDGF: platelet-derived growth factor; PDE4: phosphodiesterase 4; PPAR: peroxisome proliferator-activated receptor; PTP1B: protein tyrosine phosphatase 1B; PXR: pregnane-X-receptor; ROS: reactive oxygen species; SERCA2A: SR Ca^2+^ ATPase 2A; TLR5: Toll-like receptor 5; TDP-43: trans-activation response DNA-binding protein of 43 kDa; UVB: ultraviolet B; V-ATPase: vacuolar-type H^+^-ATPase; VEGF: vascular endothelial growth factor.

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
