# Peer review of "Marine Pharmacology in 2014–2015: Marine Compounds with Antibacterial, Antidiabetic, Antifungal, Anti-Inflammatory, Antiprotozoal, Antituberculosis, Antiviral, and Anthelmintic Activities; Affecting the Immune and Nervous Systems, and Other Miscellaneous Mechanisms of Action"

_marinedrugs, 2019, doi:10.3390/md18010005_

Round 1
Reviewer 1 Report
The authors are experts in the field and as in their previous reports they have mastered a gigantic compilation of the relevant literature.
The style/figures/tables are along the lines of the previous reports and are excellent.
We are all appreciating such reviews and we thank the authors for the hard work.
Reviewer 2 Report
Referee report to the manuscript entitled:
Marine Pharmacology in 2014–2015: Marine Compounds with Antibacterial, Antidiabetic, Antifungal, Anti-Inflammatory, Antiprotozoal, Antituberculosis, Antiviral and Anthelmintic Activities; Affecting the Immune and Nervous Systems, and other Miscellaneous Mechanisms of Action
By:
Alejandro M. S. Mayer, Aimee J. Guerrero, Abimael D. Rodríguez, Orazio Taglialatela-Scafati, Fumiaki Nakamura and Nobuhiro Fusetani
This review is written by the leading experts in the field and covers all aspects in the subject marine pharmacology. The subject is comprehensively covered. Impressive.
Small suggestions:
+) Please write in the references always also the umlauts – like König for Konig.
+) Please: no capitals in the title of the references: Antiviral Limonoids Including Khayanolides from the Trang Mangrove Plant Xylocarpus moluccensis.
+) Please: also make it uniform: Environ. Toxicol. Pharmacol. 2014, 37, (3), 1090-1100. No need to include the issue number.
It is an outstanding great review – a masterpiece.
